# Revisiting Design Choices in Offline Model-Based Reinforcement Learning

**Cong Lu**$^*$, **Philip J. Ball**$^*$, **Jack Parker-Holder, Michael A. Osborne, Stephen J. Roberts**
Department of Engineering
University of Oxford

## Abstract

Offline reinforcement learning enables agents to leverage large pre-collected datasets of environment transitions to learn control policies, circumventing the need for potentially expensive or unsafe online data collection. Significant progress has been made recently in offline model-based reinforcement learning, approaches which leverage a learned dynamics model. This typically involves constructing a probabilistic model, and using the model uncertainty to penalize rewards where there is insufficient data, solving for a *pessimistic* MDP that lower bounds the true MDP. Existing methods, however, exhibit a breakdown between theory and practice, whereby pessimistic return ought to be bounded by the *total variation distance* of the model from the true dynamics, but is instead implemented through a penalty based on estimated *model uncertainty*. This has spawned a variety of uncertainty heuristics, with little to no comparison between differing approaches. In this paper, we compare these heuristics, and design novel protocols to investigate their interaction with other hyperparameters, such as the number of models, or imaginary rollout horizon. Using these insights, we show that selecting these key hyperparameters using Bayesian Optimization produces superior configurations that are vastly different to those currently used in existing hand-tuned state-of-the-art methods, and result in drastically stronger performance.

## 1 Introduction

In offline (or batch) reinforcement learning (RL) (Ernst et al., 2005; Levine et al., 2020), the goal is to leverage offline datasets of transitions in an environment to train a policy that transfers to an online task. This could have vast implications for using RL in real-world settings, as agents can make use of ever-increasing amounts of data without the need for an accurate simulator, while also avoiding expensive and potentially even unsafe exploration in the environment.

Model-based reinforcement learning (MBRL) has recently shown promise in this paradigm, obtaining state-of-the-art performance on offline RL benchmarks (Kidambi et al., 2020; Yu et al., 2021), improving upon powerful model-free approaches (e.g. Kumar et al. (2020)). MBRL works by training a dynamics model from the offline data, then optimizing a policy using imaginary rollouts from the model. This allows the agent to learn from on-policy experience, as the model is agnostic to the policy used to generate data, making it possible to achieve high returns using data collected from even a random policy. Furthermore, recent work has demonstrated the utility of world models *beyond* maximizing return, such as generalizing to unseen variations in an environment (Ball et al., 2021), transferring to new tasks (Yu et al., 2020), and learning with safety constraints (Argenson & Dulac-Arnold, 2021). Therefore, the case for MBRL in offline RL is clear: not only does it represent state-of-the-art in terms of performance, but it also provides the opportunity to maximize the signal in the offline data to generalize onto tasks beyond those encoded by the behavior policy. This is crucial for offline RL to be useful for real-world tasks (Dulac-Arnold et al., 2021), where there will inevitably be differences between the data and desired task.

However, a common failure mode of MBRL is when policies exploit the model in parts of the state-action space where the model is inaccurate. Thus, naïve application of MBRL to offline data

---

$^*$Joint first authors. Correspondence: `cong.lu@stats.ox.ac.uk, ball@robots.ox.ac.uk`

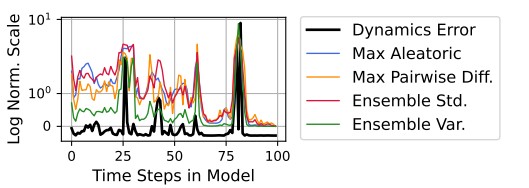
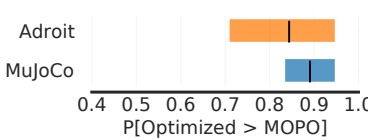

(a) **Uncertainty vs. True Errors**                    (b) **Probability of Improvement**

Figure 1: **a)** The variation of different uncertainty penalties against *true* dynamics error during a model rollout of Hopper Medium-Expert. The canonical ensemble variance penalty most closely fits the true dynamics error. **b)** Tuning key hyperparameters (an approach we call Optimized) can lead to large gains over state-of-the-art methods (MOPO) on the D4RL benchmark, as we show in this summary using `rliable` (Agarwal et al., 2021).

can result in suboptimal performance. To prevent this, concurrent works (Yu et al., 2020; Kidambi et al., 2020) have approached the problem by training a policy in a *pessimistic* MDP (P-MDP). The P-MDP lower bounds the true MDP, and discourages the policy from regions where there is large discrepancy between the true and learned dynamics; this often provides a theoretical guarantee of improvement over cloning the behavior policy that generated the offline data. This is made practically possible by adding a penalty correlated with the uncertainty in the dynamics model. However, while these recent successes are similar in principle, in practice they differ in a series of design choices. First and foremost, they make use of different heuristics to measure model uncertainty, in some cases deviating from simpler metrics which are more consistent with the theory.

In this paper, we conduct a rigorous investigation into a series of these design choices. We begin by focusing on the choice of uncertainty metric, comparing both recent state-of-the-art offline approaches (Kidambi et al., 2020; Yu et al., 2020; Rafailov et al., 2020) with additional metrics used in the online setting (Ball et al., 2020; Pan et al., 2020; Cowen-Rivers et al., 2022). We also explore the interaction with a series of other hyperparameters, such as the number of models and imaginary rollout length. Interestingly, the relationship between these variables and model uncertainty varies significantly depending on the choice of uncertainty penalty. Furthermore, we compare these uncertainty heuristics under new evaluation protocols that, for the first time, capture the specific covariate shift induced by model-based RL. This allows us to assess calibration to model exploitation in MBRL, observing that some existing penalties are surprisingly successful at capturing the errors in predicted dynamics, as seen in Fig. 1a (see App. D for details). Then, using the insights gained from sections 4 and 5, we then achieve a **43%** gain over a previously grid-searched method by using a *single hyperparameter* value across all environments. We then jointly fine-tune our identified key variables using a powerful Bayesian Optimization algorithm (Wan et al., 2021) and find the simpler uncertainty measures can provide state-of-the-art results in continuous control offline benchmarks, and that the chosen optimal hyperparameters continue to align with our analysis. Finally, we rigorously confirm the aggregate improvement of our results using the `rliable` framework (Agarwal et al., 2021) in Fig. 1b, and show that the improvements over existing methods are significant (see App. H for details). This work is intended to benefit both researchers and practitioners in offline RL. Our main findings include:

- **Longer horizon rollouts with larger penalties can improve existing methods.** Contrary to common wisdom, conducting significantly longer rollouts inside the model, coupled with larger uncertainty penalties, typically improves performance.

- **Penalties that use canonical forms of uncertainty estimation achieve better correlation with OOD measures.** The uncertainty estimation approach of Lakshminarayanan et al. (2017) often outperforms the penalty from state-of-the-art methods (Yu et al., 2020; Kidambi et al., 2020). We observe that the ensemble standard deviation is statistically strikingly similar to that used in Kidambi et al. (2020), but has improved correlation and scaling behavior.

- **Uncertainty is more correlated with *dynamics error* than *distribution shift*.** We find that successful penalties measure the discrepancy in dynamics, and can in fact assign high certainty to data far away from the offline data.

## 2    RELATED WORK

Two recent works concurrently demonstrated the effectiveness of model-based reinforcement learning (MBRL) in the offline setting. MOPO (Yu et al., 2020) follows the successful online RL algorithm MBPO (Janner et al., 2019) but trains inside a *conservative MDP*, penalizing the reward based on the maximum aleatoric uncertainty over the ensemble members. MOReL (Kidambi et al., 2020) achieves

even stronger performance, penalizing the rewards by a penalty based on the maximum pair-wise difference in ensemble member predictions. For pixel-based tasks, LOMPO (Rafailov et al., 2020) also proposed a novel penalty, using the variance of ensemble log-likelihoods. Outside the offline setting, probabilistic dynamics models leveraging uncertainty have underpinned a series of successes (Chua et al., 2018; Kurutach et al., 2018; Buckman et al., 2018; Pan et al., 2020; Pacchiano et al., 2021). Uncertainty can also be measured in MBRL without the use of neural networks (Deisenroth & Rasmussen, 2011), although these methods tend to be harder to scale and thus lack widespread use.

Effective hyperparameter selection in RL has been shown to be crucial to the success of popular algorithms (Engstrom et al., 2020; Andrychowicz et al., 2021). This becomes even more challenging in MBRL with additional hyperparameters/design-choices for the dynamics model. Recent work has shown that carefully optimizing these hyperparameters for online MBRL can significantly improve performance, with the tuned agent breaking the MuJoCo simulator (Zhang et al., 2021). In contrast, we focus on the offline setting, and investigate parameters specifically related to uncertainty estimation. Previous work studied the impact of hyperparameters in offline RL (Paine et al., 2020), finding offline RL algorithms to be brittle to hyperparameter choices. However, unlike our work they only consider model-free approaches, whereas we specifically investigate *model-based* offline algorithms. Abbas et al. (2020) investigates the impact of different uncertainty estimation methods in online MBRL; they too find penalizing with combined aleatoric and epistemic uncertainty improves performance.

Our work also relates to the rich literature on *deep ensembles* (Lakshminarayanan et al., 2017), which train multiple deep neural networks with different initializations and dataset orderings, and generally outperform variational Bayes methods (Mackay, 1992; Blundell et al., 2015). Achieving effective calibration with neural networks is notoriously difficult (Guo et al., 2017; Kuleshov et al., 2018; Maddox et al., 2019), and furthermore we require calibration under co-variate shift (Ovadia et al., 2019), as the policy learned in the model will likely deviate from the behavior policy that generated the offline data. Recent work has highlighted this issue in offline RL (Kumar et al., 2020; Yu et al., 2021) and has reported superior performance when eschewing model uncertainty entirely, and instead performing "conservative" Q-updates. However, it is unclear if this improvement is due to poor uncertainty calibration, implementation details, or a limitation in the pessimistic-MDP formulation.

## 3 BACKGROUND

All of the methods we investigate in this paper model the environment as a Markov Decision Process (MDP), defined as a tuple $M = (\mathcal{S}, \mathcal{A}, P, R, \rho_0, \gamma)$, where $\mathcal{S}$ and $\mathcal{A}$ denote the state and action spaces respectively, $P(s'|s, a)$ the transition dynamics, $R(s, a)$ the reward function, $\rho_0$ the initial state distribution, and $\gamma \in (0, 1)$ the discount factor. The goal is to optimize a policy $\pi(a|s)$ that maximizes the expected discounted return $\mathbb{E}_{\pi, P, \rho_0} \left[ \sum_{t=0}^{\infty} \gamma^t R(s_t, a_t) \right]$.

In *offline RL*, the policy is not deployed in the environment until test time. Instead, the algorithm only has access to a static dataset $\mathcal{D}_{env} = \{(s_j, a_j, r_j, s_{j+1})\}_{j=1}^{J}$, collected by one or more behavioral policies $\pi_b$. Following the notation in Yu et al. (2020) we refer to the distribution from which $\mathcal{D}_{env}$ was sampled as the *behavioral distribution*. The canonical approach in offline MBRL is to train an ensemble of $N$ probabilistic dynamics models (Nix & Weigend, 1994). These usually learn to predict both the next state $s_{t+1}$ and reward $r_t$ from a state-action pair, and are trained on $\mathcal{D}_{env}$ using supervised learning. Concretely, each of the $N$ models output a Gaussian $\widehat{P}_\phi^i(s_{t+1}, r_t|s_t, a_t) = \mathcal{N}(\mu_\phi^i(s_t, a_t), \Sigma_\phi^i(s_t, a_t))$ parameterized by $\phi$. The resulting learned dynamics model $\widehat{P}$ and reward model $\widehat{R}$ define a *model MDP* $\widehat{M} = (\mathcal{S}, \mathcal{A}, \widehat{P}, \widehat{R}, \rho_0, \gamma)$. To train the policy, we use $k$-step rollouts inside $\widehat{M}$ to generate trajectories (Sutton, 1991).

To prevent policy exploitation in a model, a pessimistic MDP (P-MDP) is constructed by lower bounding the true-expected return, $\eta_M(\pi)$, using some error between the true and estimated models. For instance, in Yu et al. (2020), the authors show that a lower bound on the return can be established by penalizing the reward by a measure that corresponds to estimated model error:

$$\eta_M(\pi) \geq \mathbb{E}_{(s,a) \sim \rho_{\hat{P}}^\pi} \left[ R(s, a) - \gamma |G_{\hat{M}}^\pi(s, a)| \right] \tag{1}$$

where $\rho_{\hat{P}}^\pi$ represents transitioning under the dynamics model $\hat{P}$ and policy $\pi$. Several potential choices for $|G_{\hat{M}}^\pi(s, a)|$ are proposed, including an upper bound based on the total variation distance

between the learned and true dynamics. However, for their practical algorithm, the authors elect to use a heuristic based on impressive empirical results. Concurrent to MOPO, MOReL (Kidambi et al., 2020) in theory constructs a P-MDP by augmenting a standard MDP with a negative valued absorbing state that is transitioned to when total variation distance between true and learned dynamics is exceeded. They show that a policy learned in this P-MDP exceeds simple behavior cloning. However, while dynamics-based total variation distance has desirable theoretical properties, the practical algorithm relies on another heuristic to approximate this quantity. This motivates the study of penalties used, as well as other under-used candidates, and their overall effectiveness.

## 4 UNCERTAINTY PENALTY

The key idea underpinning recent success in offline MBRL is the introduction of a P-MDP, penalized by some uncertainty penalty. The theory dictates this should be some distance measure between the true and predicted dynamics. Of course, this cannot be truly estimated without access to an oracle, so a proxy for this quantity is constructed instead based on uncertainty heuristics. In this paper, we compare the following uncertainty heuristics, from recent works in both offline and online MBRL:

**Max Aleatoric (Yu et al., 2020):** $\max_{i=1,\ldots,N}||\Sigma_\phi^i(s,a)||_F$, which corresponds to the maximum aleatoric error, computed over the variance heads of the model ensemble.

**Max Pairwise Diff (Kidambi et al., 2020):** $\max_{i,j}||\mu_\phi^i(s,a) - \mu_\phi^j(s,a)||_2$, which corresponds to the pairwise maximum difference of the ensemble predictions.

**LL Var (Log-Likelihood Variance) (Rafailov et al., 2020)):** $\mathrm{Var}(\{\log \widehat{P}_\phi^i(s'|s,a), i=1,\ldots,N\})$, where $s'$ is a next state sampled from a single ensemble member. We evaluate its log-likelihood under each ensemble member and take the variance.

**LOO KL (Leave-One-Out KL Divergence (Pan et al., 2020):** $D_{\mathrm{KL}}[\widehat{P}_{\phi_i}(\cdot|s,a)||\widehat{P}_{\phi_{-i}}(\cdot|s,a)]$, which corresponds to the KL divergence between the Gaussian parameterized by a randomly selected ensemble member, and the aggregated Gaussian of the remaining ensemble members.

**Ensemble Standard Deviation/Variance (Lakshminarayanan et al., 2017):** The variance is given as: $\Sigma^*(s,a) = \frac{1}{N}\sum_i^N((\Sigma_\phi^i(s,a))^2 + (\mu_\phi^i(s,a))^2) - (\mu^*(s,a))^2$ where $\mu^*$ is the mean of the means $(\mu^*(s,a) = \frac{1}{N}\sum_i^N \mu_\phi^i(s,a))$. This corresponds to a combination of epistemic and aleatoric model uncertainty. This is surprisingly under-utilized in offline MBRL, and is a canonical method of uncertainty estimation used in the Bayesian inference literature (Ovadia et al., 2019; Filos et al., 2019; Scalia et al., 2020). We choose to evaluate both standard deviation (the square root of the above) and variance, as this will provide intuition about the importance of penalty distribution *shape*.

These can all be computed using the output from an ensemble of probabilistic dynamics models (Lakshminarayanan et al., 2017), so we are able to compare them in a controlled manner.

### 4.1 HOW WELL DO ENSEMBLE PENALTIES DETECT OUT OF DISTRIBUTION ERRORS?

We begin by assessing how well uncertainty penalties correlate with next state MSE (we justify the MSE under deterministic dynamics in App. A.2). This is crucial in penalizing the policy from visiting parts of the state-action space where the model is inaccurate, and therefore exploitable. Using D4RL (Fu et al., 2021a), we train models on each dataset, then evaluate them on *other datasets* from the same environment, but collected under *different* policies. These form our "Transfer" experiments as they directly measure the ability of uncertainty penalties at detecting errors on *unseen* data. We compare the penalties against true MSE for a variety of settings in App. A.3, and summarize this in the "Transfer" column of Table 1. We measure Spearman rank ($\rho$) and Pearson bivariate ($r$) correlations, and justify their use in App. A.1. Full details of all experiments and hyperparameters are given in App. G. We will analyze these results in detail in the next section, after introducing a novel protocol for assessing our penalties under the out-of-distribution (OOD) data induced by model exploitation.

### 4.2 HOW DO THESE PERFORM DURING AN IMAGINARY ROLLOUT?

We additionally design an experiment aimed at capturing the OOD data *generated by the actual offline MBRL process*, which we call our "True Model-Based" experiments. First, we train a set of policies with 4 different starting seeds *without* a penalty inside the model for 500 iterations. We

Table 1: Correlation statistics of penalties against true mean-sq. model error, averaged over all datasets (i.e., Random through to Expert) showing ± 1 SD over 12 seeds. The best in each column is **bolded**. The ensemble penalties generally perform best.

| | Transfer | | | | True Model-Based | | | |
| | HalfCheetah | | Hopper | | HalfCheetah | | Hopper | |
| Penalty | $\rho$ | r | $\rho$ | r | $\rho$ | r | $\rho$ | r |
|---|---|---|---|---|---|---|---|---|
| Max Aleatoric | 0.78±0.00 | 0.55±0.01 | 0.71±0.01 | 0.41±0.01 | 0.58±0.01 | 0.42±0.01 | 0.73±0.03 | 0.48±0.01 |
| Max Pairwise Diff. | 0.79±0.01 | 0.62±0.00 | 0.77±0.00 | 0.57±0.00 | 0.58±0.01 | **0.52**±0.01 | 0.75±0.02 | **0.55**±0.02 |
| Ens. Std. | **0.82**±0.01 | 0.64±0.01 | **0.79**±0.00 | 0.56±0.00 | **0.61**±0.01 | **0.52**±0.00 | **0.79**±0.02 | **0.55**±0.02 |
| Ens. Var. | **0.82**±0.01 | **0.67**±0.00 | **0.79**±0.00 | **0.59**±0.00 | 0.60±0.01 | 0.49±0.01 | 0.77±0.02 | **0.55**±0.02 |
| LL Var. | 0.13±0.05 | 0.14±0.02 | 0.36±0.04 | 0.12±0.02 | 0.04±0.16 | 0.07±0.06 | 0.50±0.02 | 0.16±0.02 |
| LOO KL | 0.03±0.02 | 0.11±0.02 | 0.11±0.02 | 0.08±0.02 | -0.02±0.12 | 0.06±0.06 | 0.22±0.02 | 0.10±0.02 |

then measure the difference between the return predicted by the model over a rollout, and the true return in the real environment. We define a policy to be "exploitative" if the model significantly *over-estimates* the return compared to the true return. It is these exploitative policies that induce the types of extrapolation errors which cause MBRL methods to fail in the offline setting. It is therefore important that the penalty is able to accurately determine when the model is being exploited in this way. We use a subset of the 5 most exploitative policies to generate trajectories in the model, and record the uncertainty predicted by each penalty at every time step. To generate the True Model-Based data, we then "replay" these trajectories in the true environment, loading the state and action taken in the model into the environment, and record the "true" next state according to the MuJoCo simulator (Todorov et al., 2012). True Model-Based therefore calculates the MSE between the *predicted* and *actual* next states. Table 1 summarizes the results from both the Transfer and True Model-Based experiments. Additional details are provided in App. D along with full correlation plots in App. A.3.

We are now in a position to analyze the results in Table 1. It is immediately obvious that the LOO KL and LL Var penalties have very weak correlation with MSE. We believe this is because LL Var relies on likelihood statistics, which are notoriously sensitive; it was designed for use with a KL-regularized latent state space model which has well-behaved dynamics. Regarding LOO KL, we note that this penalty was designed for the online setting with significantly less data, and becomes quite uncorrelated in this larger data setting. This advocates penalties that are less reliant on distributional information concerning the separate Gaussians in the ensemble, as such penalties appear sensitive to the quality of their estimated distributions. We observe that Max Aleatoric, Max Pairwise Diff and the Ensemble penalties perform broadly similarly despite their different analytical forms; Ensemble measures do however exhibit noticeably higher rank correlation. We also observe a significant performance loss between the Transfer and True Model-Based HalfCheetah settings, with the latter being relatively poor. This implies further work is needed to develop penalties that can successfully detect the type of dynamics discrepancies that actually arise in offline MBRL. Finally, we observe that despite the similar rank correlations $\rho$, the bivariate correlations $r$ can vary considerably, and observe from the scatter plots that Max Aleatoric exhibits low kurtosis, having large penalty values "bunched" at its extreme; we provide 3rd and 4th order moment statistics to facilitate shape comparisons in App. C.

## 5    KEY HYPERPARAMETERS IN OFFLINE MBRL

### 5.1    HOW MANY MODELS DO WE NEED?

At present the number of models used has not been discussed since MBPO, which trains seven probabilistic dynamics models of the same architecture (with different initializations), using only the top five models based on validation accuracy (referred to as "Elites" in the Evolutionary community, e.g. Mouret & Clune (2015)). The reason or justification for this is not discussed, but it has seemingly been adopted in the wider MBRL setting (Shen et al., 2020; Omer et al., 2021; Pineda et al., 2021). However, offline RL is a totally different paradigm, where it is possible that access to compute is less of a bottleneck and it may be preferable to use more models to extract the most signal possible from the static dataset. Inevitably, many of the ensemble penalties are dependent on the number of models; for example, it is easy to see that the Max Aleatoric value could scale poorly with more models.

**How Does Penalty Distribution Change with Model Count?** We now vary the number of models used in the calculation of the penalties and plot their respective distributions; an illustrative example is shown in Fig. 2 with full results in App. B. The scaling of penalties relying on max over sets is most affected with increasing the number of models due to admitting more extreme values, and we observe that the distribution shape of Max Aleatoric changes significantly as we admit more models, which

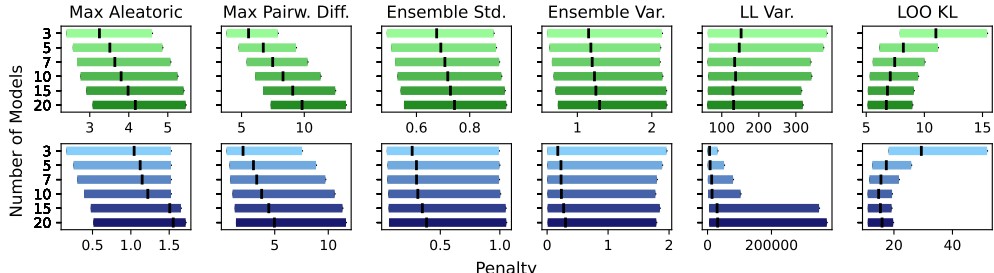

Figure 2: Box Plots showing D4RL Medium transferred to Random. We show IQR limits and the median value denoted by the black vertical line. Green = HalfCheetah, Blue = Hopper. Max Aleatoric, Max Pairw. Diff. and LOO KL are unstable w.r.t. ensemble member count. In contrast, ensemble variance and std. are far more stable.

we validate in App. C. This impacts the tuning of this hyperparameter, as we have to contend with a changing distribution along with calibration quality (which we explore in the next section). Finally, we observe that the Ensemble penalties change the least with differing model count, highlighting their ease of tuning; this is clearly a desirable property for designing such metrics going forward.

**How does Penalty Performance Scale with Model Count?** Empirically, there exists an optimal number of models to use in an ensemble for model-based RL (Kurutach et al., 2018; Matsushima et al., 2021). Up to now, heuristics have been used to select how many models we use for uncertainty estimation, despite it being possible to use a different number of models for dynamics prediction and uncertainty estimation. For instance, in MOPO, transitions are generated with five Elite models, but all seven models are used to calculate the penalty. In MOReL, four models are

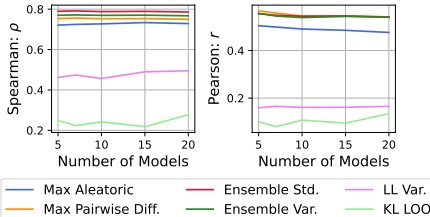

Figure 3: Plot of how error and penalty correlation changes with model number in Hopper across all datasets (i.e., Random through to Expert).

used for both transitions and penalty prediction. Therefore, we wish to understand if there is merit to using a larger number of models for uncertainty estimation compared with next state prediction. We provide a snapshot in Fig. 3, showing the aggregated results on the True Model-Based data in Hopper, with full results in App. B. We see there is no clear consensus, and that the optimal number of models is highly dependent environment, the behavior data, and penalty type, with some settings showing improved calibration with model count and vice-versa. This clearly justifies treating the number of models as a hyperparameter that is important to tune, especially on transfer tasks. Interestingly, we observe that it is possible to simultaneously improve rank ($\rho$) correlation, but reduce bivariate ($r$) correlation, especially with the MOPO penalty. This again suggests that the number of models not only affects the quality of the estimation, but *also its distributional shape*.

## 5.2 THE WEIGHT OF UNCERTAINTY $\lambda$

To weight penalty against reward, MOPO introduces a parameter $\lambda$ that trades off between the two terms. In their paper, the authors sweep over $\lambda \in \{1, 5\}$ for each environment. However, the optimal values may lie outside this region. Furthermore, we have shown this value will need to drastically change to account for using a different penalty or even number of models.

## 5.3 THE ROLLOUT HORIZON $h$

The horizon $h$ of the rollouts plays a crucial role in offline RL. Longer horizon rollouts increase the likelihood of errors in the transitions (we verify this intuition in App. D), but conversely can improve performance when errors are properly managed (Janner et al., 2019; Pan et al., 2020). Furthermore, as highlighted in Fig. 1a, the model can generalize, and dynamics error does not necessarily increase with drift away from the offline dataset. Instead, we observe spikes, and note it is possible to recover from these to valid states and transitions. It is therefore imperative that a penalty captures these spikes over the course of an entire model rollout with horizon $h$, and down-weights the reward accordingly.

Using this observation, we design a novel experiment that treats these spikes as "positive" labels, and normalize each penalty to $[0, 1]$. This converts the penalties into a probabilistic classifier, and we evaluate how well they classify these events that occur increasingly under longer $h$. This is precisely the intuition behind the LOO KL and LL Var approaches, whereby the penalty acts as an

Table 2: Performance of different penalties as OOD event detectors averaged over all datasets in Hopper and HalfCheetah (i.e., Random through to Expert) showing $\pm$ 1 SD over 12 seeds. AUC is "Area Under Curve" and AP is "Average Precision". The best (highest) in each column is highlighted in **bold**.

| | Percentile | | | | | | | | | | | |
| --- | --- | --- | --- | --- | --- | --- | --- | --- | --- | --- | --- | --- |
| | 90th | | | | 95th | | | | 99th | | | |
| | Dynamics | | Distribution | | Dynamics | | Distribution | | Dynamics | | Distribution | |
| Penalty | AUC | AP | AUC | AP | AUC | AP | AUC | AP | AUC | AP | AUC | AP |
| Max Aleatoric | 0.89±0.01 | 0.50±0.02 | 0.76±0.01 | 0.35±0.01 | 0.89±0.00 | 0.35±0.02 | 0.80±0.01 | 0.27±0.01 | 0.92±0.00 | 0.20±0.04 | 0.89±0.03 | 0.16±0.04 |
| Max Pairwise Diff. | **0.90**±0.00 | 0.54±0.01 | 0.77±0.01 | 0.34±0.01 | **0.91**±0.00 | 0.40±0.02 | 0.81±0.01 | 0.28±0.01 | **0.93**±0.00 | 0.26±0.01 | 0.89±0.02 | 0.15±0.02 |
| Ensemble Std. | **0.90**±0.00 | 0.55±0.01 | **0.79**±0.01 | **0.38**±0.01 | **0.91**±0.00 | 0.40±0.02 | **0.83**±0.01 | **0.31**±0.01 | **0.93**±0.00 | 0.25±0.02 | **0.90**±0.02 | **0.18**±0.02 |
| Ensemble Var. | **0.90**±0.00 | **0.56**±0.01 | 0.78±0.01 | 0.35±0.01 | **0.91**±0.00 | **0.42**±0.02 | 0.82±0.01 | 0.29±0.01 | **0.93**±0.00 | **0.27**±0.01 | 0.89±0.02 | 0.16±0.02 |
| LL Var. | 0.66±0.03 | 0.33±0.00 | 0.74±0.02 | 0.33±0.00 | 0.67±0.02 | 0.21±0.02 | 0.76±0.02 | 0.25±0.02 | 0.73±0.03 | 0.09±0.01 | 0.81±0.02 | 0.11±0.01 |
| LOO KL | 0.59±0.03 | 0.21±0.01 | 0.68±0.00 | 0.24±0.02 | 0.60±0.02 | 0.12±0.00 | 0.70±0.01 | 0.14±0.02 | 0.65±0.03 | 0.04±0.00 | 0.72±0.02 | 0.05±0.00 |

anomaly detector, removing detrimental transitions that lie above a threshold. This is the regime we focus on here, where binary detection is more important than correlation. Finally, we assess two "True Model-Based" errors: the dynamics error as before, and introduce the distance from the offline distribution trained on, which we calculate as the 2-norm between a state-action tuple and its nearest point in the offline data (Dadashi et al., 2021); these are called "Dynamics" and "Distribution" respectively. We provide precision-recall curves and more details on this experiment in App. D and E.

We observe in Table 2 that the penalties are powerful at identifying dynamics discrepancy, but not as accurate at identifying when the world-model data is out-of-distribution with respect to the offline data. This is a well-known phenomenon in deep neural networks and has been recently investigated in terms of feature collapse (Van Amersfoort et al., 2020), where latent representations of points far away in the input space get mapped close together. On the other hand, this shows an important distinction between the regularization induced by MBRL uncertainty and explicit state-action regularization in model-free approaches, such as Kumar et al. (2020); Wu et al. (2021). In the latter approaches, policies are penalized for taking out of distribution actions w.r.t. the offline dataset, but this is not always the case with policies trained under MBRL and uncertainty penalties. The success of MBRL methods in RL may therefore lie in the generation of state-action samples that are **OOD but represent accurate dynamics**, thus facilitating dynamics generalization in policies; recent work has shown that augmenting dynamics improves offline RL policy generalization (Ball et al., 2021). We believe future work understanding the implications of this property is vitally important.

## 6 TESTING THE LIMITS OF CURRENT APPROACHES

Given our previous analysis, in this section we seek to answer the following question: how well can existing methods perform with a more optimal selection of the discussed hyperparameters? To answer this, we consider, first, a naïve selection of one hyperparameter set across all environments (based on our previous analysis), and then more definitively, tuning the configuration for each individual D4RL MuJoCo environment using a state-of-the-art Bayesian Optimization (BO) algorithm (Wan et al., 2021). Our first set of results show that following our analysis can provide significant gains over existing baselines, whilst the second beats the current SoTA. Note, previous analysis focused on HalfCheetah and Hopper environments, so we extend our evaluation to Walker2d as a held-out test.

**General applicability of our insights.** Two of our main takeaways in Sections 4 and 5 are that we should favor the canonical Ensemble penalties and longer rollout horizons. To test these claims, we design an experiment where we fix $h = 20$ for the horizon (c.f. $h = 5$ in MOPO at most), and only use Ensemble Std. as our penalty (see App. G for details). Since tuning the penalty weight $\lambda$ per environment is unrealistic, we employ an automatic penalty tuning scheme, analogous to the automatic entropy tuning used in Haarnoja et al. (2018). We tune the penalty weight on-the-fly to a constraint value of $\Lambda = 1$, meaning we *use only a single hyperparameter across all environments*. Full details on the penalty weight tuning are provided in App. I. With this approach, we get an average reward of **49.0** in the D4RL locomotion test suite (Fu et al., 2021a), an increase of **43%** over MOPO, which was grid-searched per environment.

This clearly shows that applying the findings from our analysis provides large performance gains generally. This result is the best we know for a single hyperparameter setup, and is particularly significant as other offline MBRL algorithms tune many hyperparameters per environment. This 'zero-shot' hyperparameter restriction is also the most realistic application of offline RL to real world problems. If we were to allow ourselves to take the maximum over just 2 hyperparameter setups (the second setup being $h = 10$, $\Lambda = 0.5$), we achieve an average reward of **57.8**, an increase of **69%** over MOPO. We show the full results in Table 8 in App. I with improvement probabilities.

Table 3: Best hyperparameters discovered by our BO algorithm, followed by a comparative evaluation on the D4RL benchmark suite against other model-based RL algorithms. We use D4RL v0 datasets. The raw score for Optimized[†] and MOPO[†] was taken to be the average over the last 10 iterations of policy training, averaged over 4 seeds and showing $\pm 1$ SD. Results of MOPO and COMBO were taken from the COMBO paper. Results for MOReL were taken from its paper. $\star$ indicates $p < 0.05$ for Welch's t-test for gain over MOPO. [†]Run on our codebase. [‡]Authors' reported scores. [°]Authors used D4RL v2, which has more performant offline data.

| Environment | | Discovered Hyperparameters | | | | Optimized[†] | MOPO[†] | MOPO[‡] | MOReL[°] | COMBO |
|---|---|---|---|---|---|---|---|---|---|---|
| | | N | $\lambda$ | h | Penalty | | | | | |
| HalfCheetah | random | 10 | 6.64 | 12 | Ensemble Std | 31.7 ±1.5 | 32.7 ±1.7 | 35.4 | 25.6 | **38.8** |
| | mixed | 11 | 0.96 | 37 | Ensemble Var | **58.0** ±2.5 | 52.8 ±1.1 | 53.1 | 40.2 | 55.1 |
| | medium | 12 | 5.92 | 6 | Ensemble Var | 45.7 ±2.6 | 46.5 ±0.7 | 42.3 | 42.1 | **54.2** |
| | med.-exp. | 7 | 4.56 | 5 | Max Aleatoric | **104.2** ±5.7 $\star$ | 67.6 ±23.6 | 63.3 | 53.3 | 90.0 |
| Hopper | random | 6 | 4.46 | 47 | Ensemble Std | 12.1 ±0.2 $\star$ | 4.2 ±1.5 | 11.7 | **53.6** | 17.8 |
| | mixed | 7 | 5.90 | 5 | Max Aleatoric | 90.8 ±11.1 $\star$ | 66.7 ±27.8 | 67.5 | **93.6** | 73.1 |
| | medium | 7 | 37.28 | 42 | Ensemble Std | 69.3 ±15.2 $\star$ | 17.3 ±6.3 | 28.0 | **95.4** | 94.9 |
| | med.-exp. | 12 | 39.08 | 43 | Max Aleatoric | 105.8 ±1.2 $\star$ | 24.9 ±5.5 | 23.7 | 108.7 | **111.1** |
| Walker2d | random | 10 | 0.21 | 12 | Ensemble Var | 21.7 ±0.1 $\star$ | 13.6 ±1.4 | 13.6 | **37.3** | 7.0 |
| | mixed | 13 | 2.48 | 47 | Ensemble Std | **65.8** ±17.4 $\star$ | 37.6 ±20.6 | 39.0 | 49.8 | 56.0 |
| | medium | 8 | 5.28 | 14 | Ensemble Std | **79.7** ±2.3 $\star$ | -0.1 ±0.0 | 17.8 | 77.8 | 75.5 |
| | med.-exp. | 12 | 0.99 | 37 | Ensemble Std | **97.1** ±4.9 $\star$ | 46.2 ±27.0 | 44.6 | 95.6 | 96.1 |
| Average Score | | - | - | - | - | **65.2** ±5.4 $\star$ | 34.2 ±9.8 | 36.7 | 64.4 | 64.1 |

**Testing the limits of current approaches.** Next, we wish to further validate that our earlier theoretical analysis can correspond to strong empirical performance gains by performing BO over the key hyperparameters. Details on the BO algorithm are listed in App. G. We define our search space over hyperparameters most related to *uncertainty quantification*:

- **Penalty type (categorical):** taking values over {Max Aleatoric, Max Pairwise Diff, LOO KL, LL Var, Ensemble Std, Ensemble Variance}.

- **Penalty scale $\lambda$ (continuous):** taking values over $[1, 100]$.

- **h (integer):** taking values over $\{1, 2, \ldots, 50\}$.

- **Models N (integer):** taking values over $\{1, 2, \ldots, 15\}$.

Table 3 shows the optimal hyperparameters under BO. We note that Ensemble penalties are mainly selected, corroborating the findings in our analysis that these are most correlated with model error. We observe that Max Pairwise Diff is not chosen, likely because ensemble penalties are better correlated with true dynamics error, and are more stable under tuning since their scaling changes less with model number; we know that Max Pairwise Diff has very similar shape statistics to Ensemble Std. (App. C). Finally, we also observe these solutions have lower performance variance than MOPO.

The selection of Max Aleatoric is also explainable; we observe it displays significantly lower skew and kurtosis than all other metrics (App. C), while still maintaining strong rank correlation. We also found that in all Hopper experiments, Ensemble Var. never achieved high performance, despite the only difference with Ensemble Std. being its distributional shape. Interestingly, in HalfCheetah, the opposite is true, with Ensemble Var. delivering significant performance gains. This implies that distributional shape may play as important a role as calibration, and advocates the learning of *meta-parameters* that control for this. Finally, in Walker2d, the well-grounded ensemble penalties win in all cases. We note that values of the rollout horizon $h$ and penalty weight $\lambda$ differ greatly from those chosen in the original MOPO paper, which chooses both from $\{1, 5\}$. Notably, the Hopper and Walker2d environments can prefer a much longer rollout length and higher penalty weight, even accounting for penalty magnitudes. Again this is backed up by our analysis; along a single rollout, dynamics errors do not necessarily accumulate, they simply become more likely to occur. Therefore, as long as we penalize errors appropriately, we can handle longer rollouts and, as a result, generate more on-policy data. The number of models used to compute the uncertainty estimates can also differ greatly from the standard 7. This again aligns with our findings that using more models for uncertainty estimation can be beneficial, but is dependent on environment, data, and penalty.

Table 3 also demonstrates how these unconventional hyperparameter choices fare against state-of-the-art offline MBRL algorithms. We spent considerable effort ensuring that our implementation of MOPO matched the authors' results using the same hyperparameters. We note the two are very similar[1], thereby allowing us to make a faithful comparison when modifying hyperparameters. Our approach, labeled "Optimized[†]", achieves *statistically significant* improvements over MOPO on 9

---

[1]There was a disparity in Walker2d-medium, but this was also noted in Ball et al. (2021)

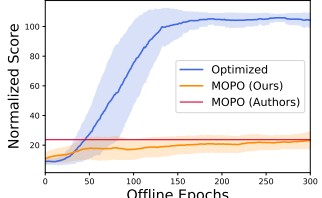

Figure 4: MOPO performance on the Hopper medium-expert environment.

Table 4: Comparative evaluation on the D4RL Adroit v0 dataset against Model Free CQL

| Environment | | MOPO[†] | Optimized[†] | CQL |
|---|---|---|---|---|
| pen | cloned | 5.4 ±10.8 | 23.0 ±4.2 | 39.2 |
| | human | 6.2 ±7.8 | 19.0 ±7.9 | 37.5 |
| | expert | 15.1 ±9.7 | 50.6 ±10.5 | 107.0 |
| hammer | cloned | 0.2 ±0.1 | 5.2 ±1.5 | 2.1 |
| | human | 0.2 ±0.0 | 0.5 ±0.8 | 4.4 |
| | expert | 6.2 ±8.4 | 23.3 ±4.1 | 86.7 |

out of 12 environments, validating our prior analysis over the key design choices. As an additional bonus, and it is not the stated aim of this work, our approach achieves state-of-the-art performance on five HalfCheetah and Walker2d environments by a considerable margin. Further notable results include the Hopper mixed and Hopper medium-expert environments, in which we show we are able to tune the MOPO-like method up to the performance of COMBO (Yu et al., 2021) and MOReL. The importance of good uncertainty estimation and hyperparameter selection is shown visually in Fig. 4 where we improve MOPO performance by over $5\times$ whilst obtaining a stable solution.

As aforementioned, we found our policies are more stable than previous works and consequently *do not need to cherry-pick* high performing checkpoints[2]. Instead, we report the average performance over our final 10 policy-improvement iterations. It should be noted that stability during training (Chan et al., 2020) is paramount for successful policy deployment in offline RL, and we should therefore prioritize hyperparameters that ensure this. We further confirm the reliability of our evaluation using the `rliable` framework (Agarwal et al., 2021) in Fig. 1b, showing that the improvement over MOPO (with 95% bootstrap CIs shaded) is clear in both MuJoCo and Adroit.

**Results on Adroit dexterous hand manipulation tasks.** We present results in Table 4 on the Adroit Pen and Hammer environments which, as far as we are aware, *have not previously been used in offline MBRL*, and present very different challenges to the locomotion tasks. These tasks feature sparse rewards, real human demonstrations and narrow data distributions. We compare against the current state-of-the-art *model-free* algorithm (CQL, Kumar et al. (2020)) and find that offline MBRL can learn useful policies in the Adroit domains, providing the best performance seen so far on the hammer-cloned setting. Best found penalties and hyperparameters are listed in App. J, and mirror the findings in the locomotion experiments. We believe issues with the world model not accurately capturing sparse rewards may account for any major performance difference. Our work is therefore an important step towards bridging the gap between model-based and model-free methods for sparse reward tasks, especially in the offline setting where exploration is not possible. We define MOPO to be the best performance with the Max Aleatoric penalty, searching $\lambda, h$ in $\{1, 5\}^2$.

## 7 CONCLUSION

In this paper, we rigorously evaluated the impact of various key design choices on offline MBRL, comparing for the first time a number of different uncertainty penalties used in the literature. By proposing novel evaluation protocols, we have also gained key insights into the nature of uncertainty in offline MBRL that we believe benefits the RL community. We demonstrated the impact of this analysis by significantly improving upon existing offline MBRL by using vastly different key hyperparameters, obtaining statistically significant performance improvements in almost all benchmarks.

Going forward, we are excited by developments in offline evaluation (Chen et al., 2021; Fu et al., 2021b) to accurately assess agent performance without querying the environment. This would open the door for population-based training methods (Jaderberg et al., 2017; Parker-Holder et al., 2020), which have shown great success in online MBRL (Zhang et al., 2021). Furthermore, throughout the paper we have highlighted potential areas of interest, from better understanding the generalization provided by world models, through to the development of meta-parameters controlling penalty distribution shape. We also highlight key issues in implementation in App. F as we strongly believe this is a vital frontier for disentangling the effect of algorithmic innovations from code-level details. Finally, Offline MBRL so far has only focused on determinstic environments; the ensemble penalties we investigate support the modeling of stochastic dynamics, and the novel tools for analysis we develop here can be readily applied to such settings.

---

[2]It is unclear what procedure is used in some prior work (indeed issues have been raised about this).

ACKNOWLEDGMENTS

The authors would like to acknowledge Rishabh Agarwal for helpful feedback during the project. We would also like to thank the anonymous reviewers for their constructive feedback, which helped to improve the paper. Cong Lu is funded by the Engineering and Physical Sciences Research Council (EPSRC). Philip J. Ball is funded through the Willowgrove Studentship.

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

# A CALIBRATION

## A.1 CHOICE OF CALIBRATION METRICS

We consider both the Spearman rank ($\rho$) correlation and Pearson bivariate ($r$) correlation. We believe that the former better represents the actual statistical power of the metric compared to the true distributional shift value, as it is robust to outliers and isn't impacted by distributional shape (i.e., skewness, kurtosis). After all, we do not know if some 'true' $|G_{\hat{M}}^{\pi}(s, a)|$ is even linearly correlated with the MSE values that we report, so naïvely comparing based on bivariate correlation may result in incorrect assessment of penalty efficacy. However, we do also include the Pearson bivariate correlation to gain insight into how the penalty distribution shape changes with design choices. For instance, consider two metrics that have identical Spearman coefficients, but vastly different Pearson coefficients–this implies they have significantly different distributional shapes whilst having the same statistical ranking power. The two correlation coefficients have the further advantage that they are unaffected by the scale of the uncertainty penalty, which can vary widely. Furthermore, algorithms such as MOPO and MOReL will often scale the penalty by some coefficient $\lambda$ and thus the raw unscaled value is hard to interpret.

## A.2 THE USE OF MSE AS THE GROUND TRUTH FOR DETERMINISTIC DYNAMICS

Following Yu et al. (2020), it is possible upper bound the expected performance $\eta_M$ of a policy $\pi$ in the true MDP $M$ under training in a world model MDP $\hat{M}$ as follows:

$$\eta_M(\pi) \geq \mathop{\mathbb{E}}_{(s,a)\sim\rho_{\hat{P}}^{\pi}} \left[ R(s, a) - \gamma|G_{\hat{M}}^{\pi}(s, a)| \right] \tag{2}$$

where $R(\cdot, \cdot)$ is the reward function, $\rho_{\hat{P}}^{\pi}$ represents transitioning under the world model dynamics $\hat{P}$ and policy $\pi$. The quantity $|G_{\hat{M}}^{\pi}(s, a)|$ can be upper-bounded by an integral probability metric (IPM):

$$|G_{\hat{M}}^{\pi}(s, a)| \leq \sup_{f\in\mathcal{F}} \left| \mathbb{E}_{s'\sim\hat{P}(s,a)}[f(s')] - \mathbb{E}_{s'\sim P(s,a)}[f(s')] \right| =: d_{\mathcal{F}}(\hat{P}(s, a), P(s, a)) \tag{3}$$

where $\mathcal{F}$ is some set of functions mapping $\mathcal{S}$ to $\mathbb{R}$, and $P$ is the dynamics under the true MDP $M$. As noted in Yu et al. (2020), making assumptions over the functional form of $\mathcal{F}$ induces different distance measures. Restricting $\mathcal{F}$ to the set of 1-Lipschitz functions results in an IPM with the following form:

$$|G_{\hat{M}}^{\pi}(s, a)| \leq cW_1(\hat{P}(s, a), P(s, a)) \tag{4}$$

which is the 1-Wasserstein distance, where the constant $c$ is the Lipschitz constant of the value function $V_M^{\pi}$ with respect to a norm $||\cdot||$. Recalling that the environments we evaluate have deterministic dynamics (Todorov et al., 2012), this means the dynamics distributions $P$ and $\hat{P}$ in Eq. 4 are Dirac delta functions. In this case, the 1-Wasserstein distance simply reduces to the 2-norm between some 'true dynamics' $T(s, a)$ and the 'estimated dynamics' $\hat{T}(s, a)$. This justifies the use of MSE between the oracle dynamics (as detailed in Sec. 4.1 and App. D.1) and the world model dynamics as the *ground truth* measure under which we assess calibration.

### A.3 OFFLINE DATASET TRANSFER CALIBRATION

We present the full calibration scatter plots described in Sec. 4.2. Concretely, we plot penalty values on the y-axis, and ground-truth MSE on the x-axis. First, we present the transfer performance of training sets onto offline datasets. Then, we present the results for all training datasets under the True Model-Based experiment under the adversarial policies.

### A.3.1 HALFCHEETAH

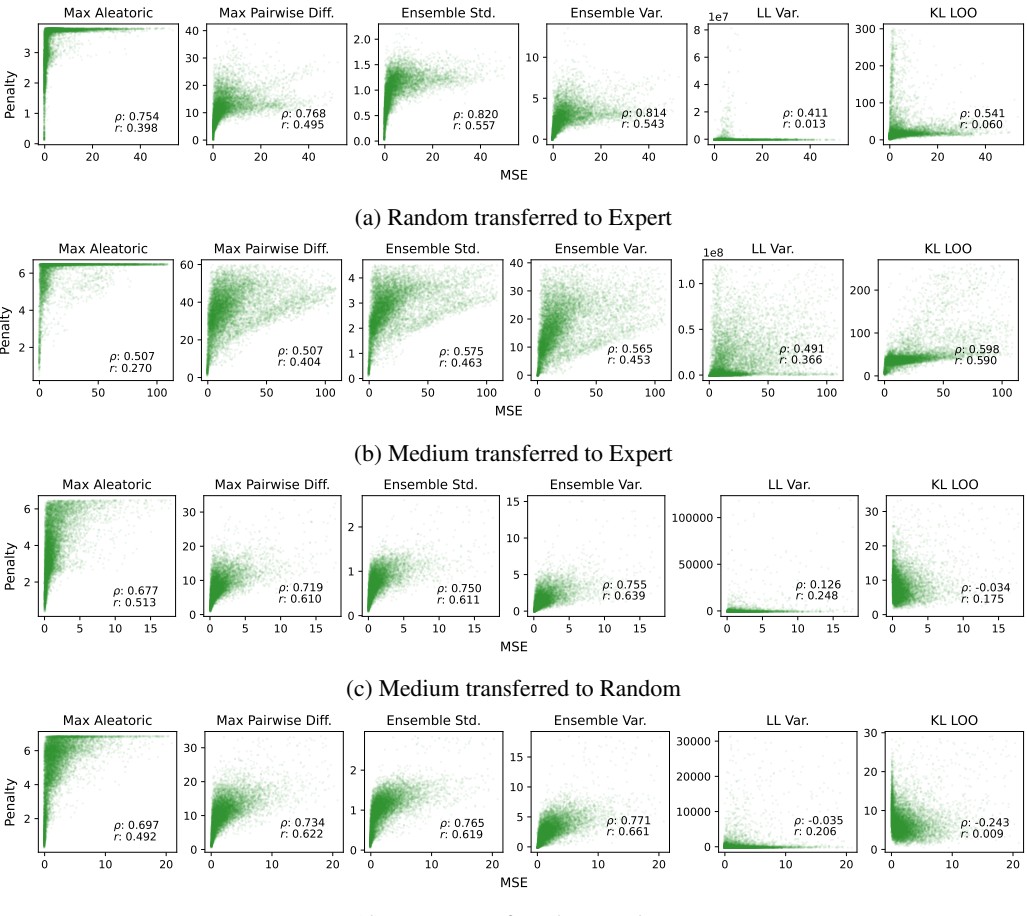

Figure 5: Scatter Plots showing HalfCheetah D4RL transfer tasks.

### A.3.2 HOPPER

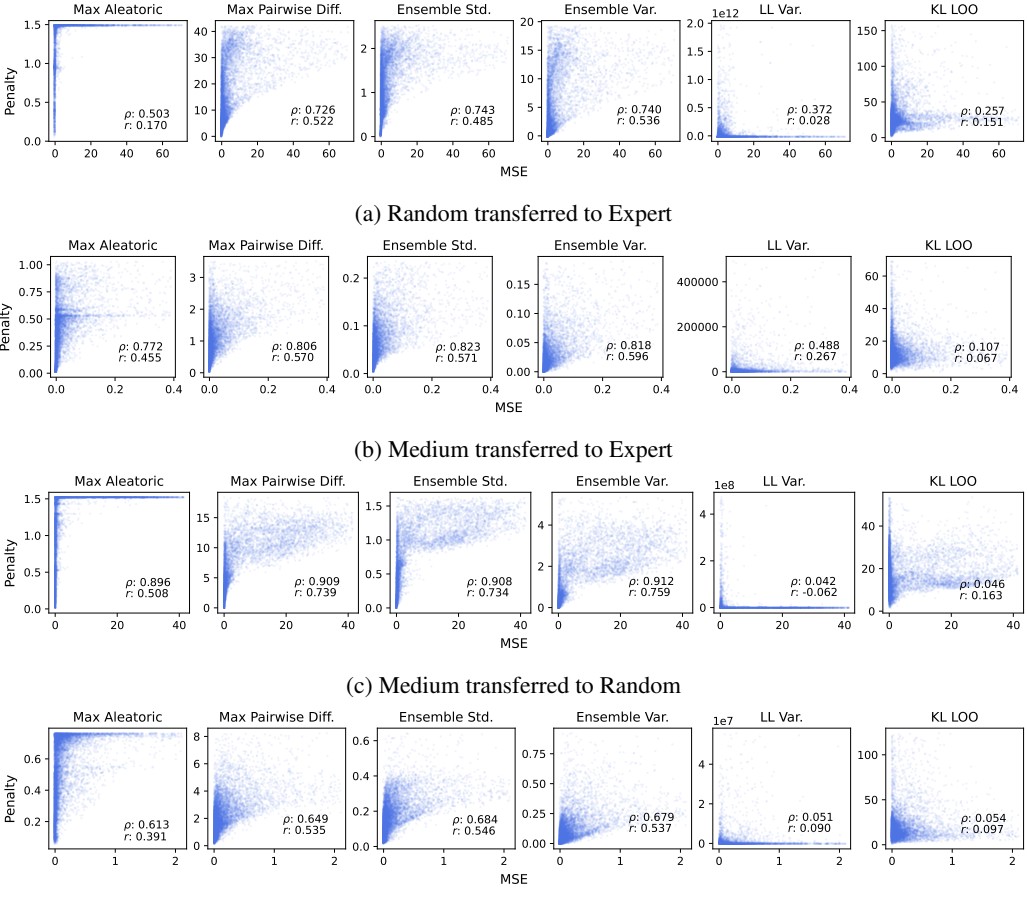

Figure 6: Scatter Plots showing Hopper D4RL transfer tasks.

## A.4 TRUE MODEL-BASED ERROR CALIBRATION

### A.4.1 HALFCHEETAH

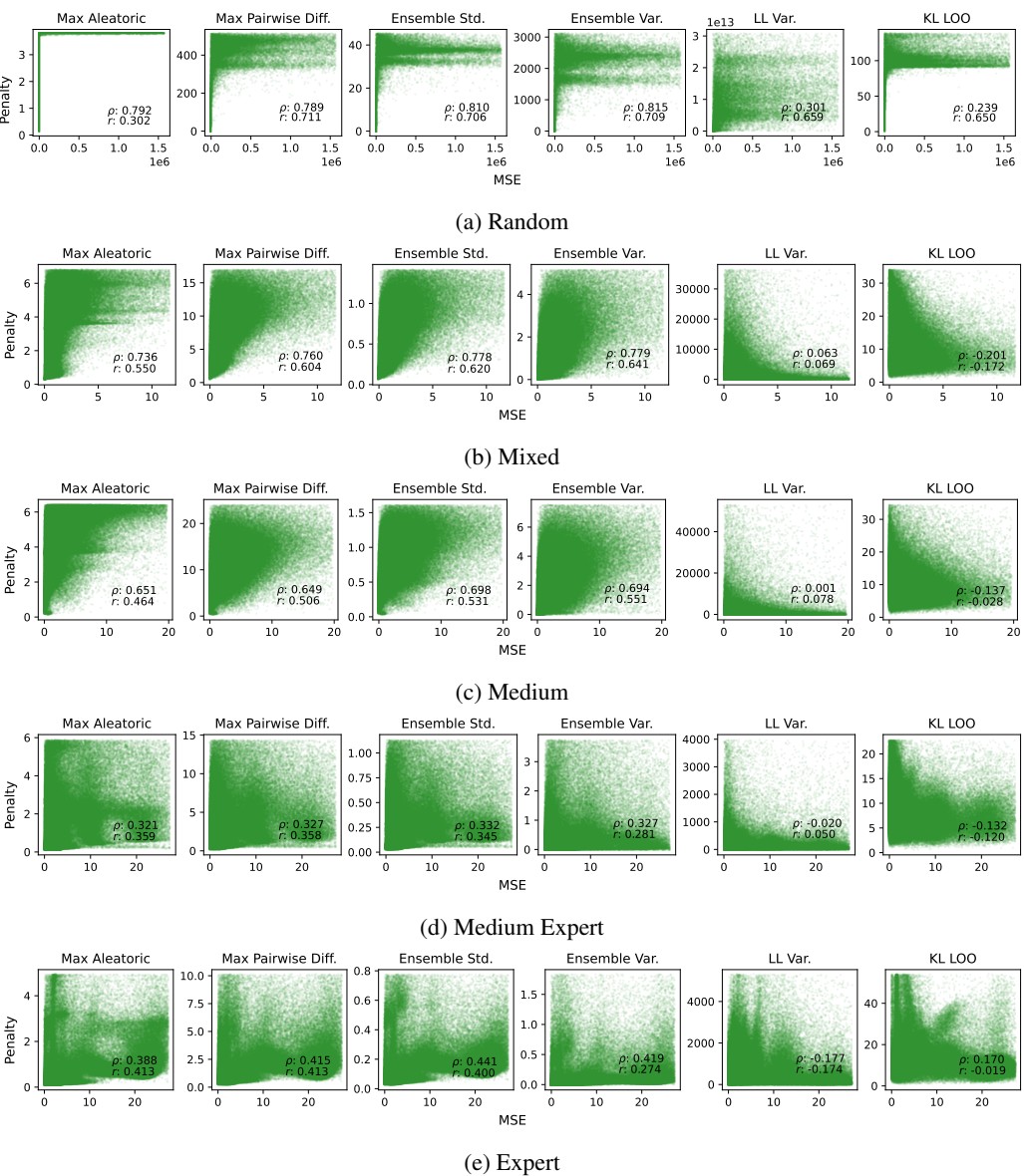

Figure 7: Scatter Plots showing HalfCheetah D4RL true model-based error calibration.

### A.4.2 HOPPER

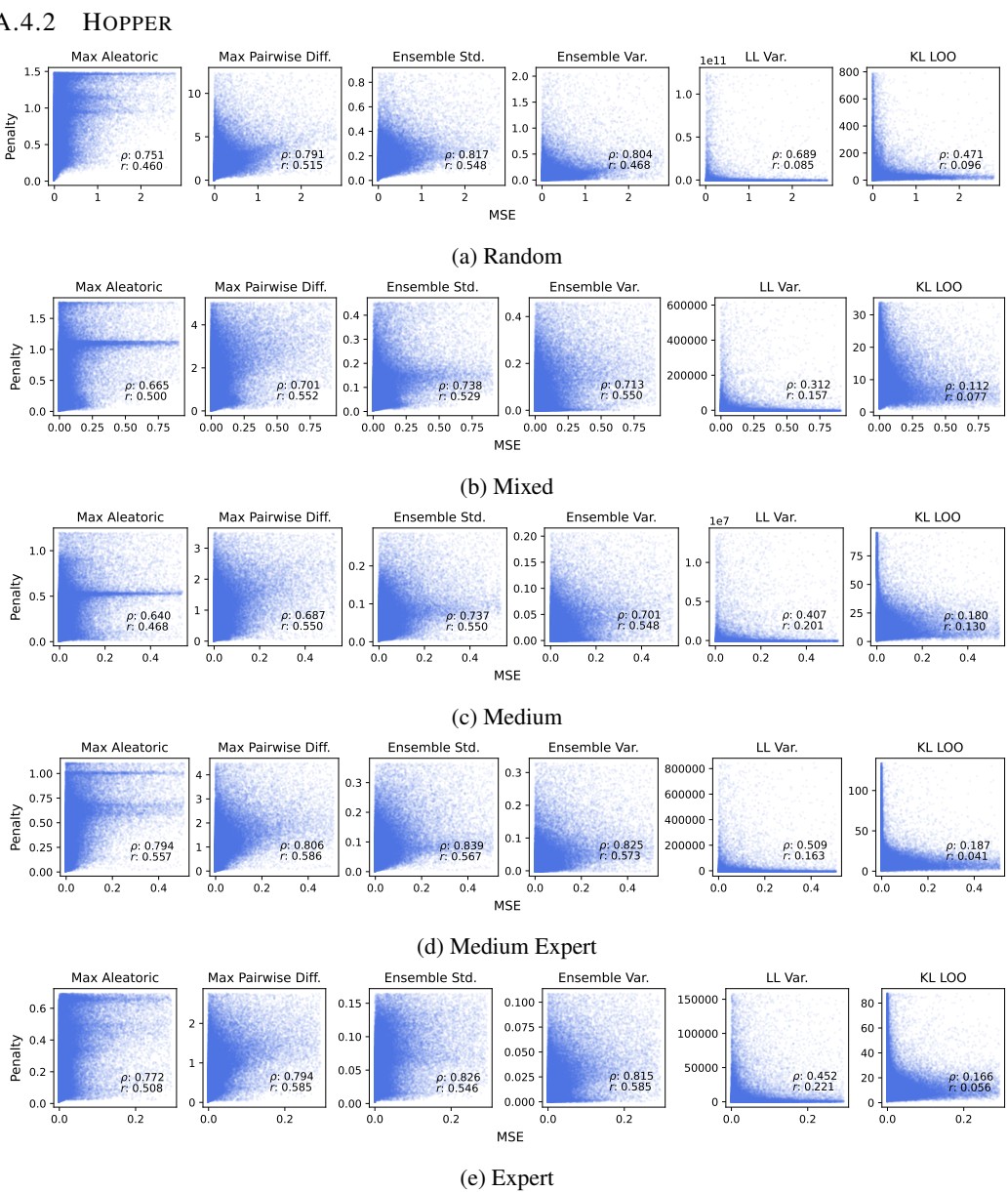

Figure 8: Scatter Plots showing Hopper D4RL true model-based error calibration.

## A.5 ADDITIONAL LOG PROBABILITY CORRELATION ANALYSIS

Table 5 shows correlation between reward penalties and the negative log-likelihood of the true data under the model in the Transfer experiments.

Table 5: Correlation statistics of penalties against model negative log-likelihood of the true data, averaged over all datasets (i.e., Random through to Expert) showing ± 1 SD over 12 seeds. The best in each column is **bolded**.

| | Transfer | | | |
|---|---|---|---|---|
| | **HalfCheetah** | | **Hopper** | |
| **Penalty** | $\rho$ | r | $\rho$ | r |
| Max Aleatoric | 0.87±0.00 | 0.82±0.01 | 0.81±0.01 | 0.57±0.01 |
| Max Pairwise Diff. | 0.79±0.01 | 0.62±0.00 | 0.79±0.01 | 0.51±0.00 |
| Ens. Std. | **0.93**±0.00 | **0.86**±0.01 | **0.89**±0.01 | **0.61**±0.01 |
| Ens. Var. | 0.90±0.01 | 0.74±0.01 | 0.82±0.00 | 0.59±0.00 |
| LL Var. | 0.04±0.07 | 0.07±0.03 | 0.25±0.03 | 0.10±0.01 |
| LOO KL | -0.04±0.06 | -0.02±0.04 | 0.08±0.03 | 0.05±0.01 |

# B  FULL RESULTS INCREASING MODELS

## B.1  PENALTY DISTRIBUTION

In this section we provide the full set of results showing the impact of increasing model count on the distribution quantile statistics as introduced in Sec. 5.1. We show inter-quartile range and the median (the latter being denoted by a black vertical line) of each penalty as a function of increasing model number across all training domains and test settings. First, we present the transfer performance of all training sets onto all offline datasets. Then, we present the results for all training datasets under the True Model-Based experiment under the adversarial policies.

### B.1.1  OFFLINE DATASET TRANSFER DISTRIBUTION

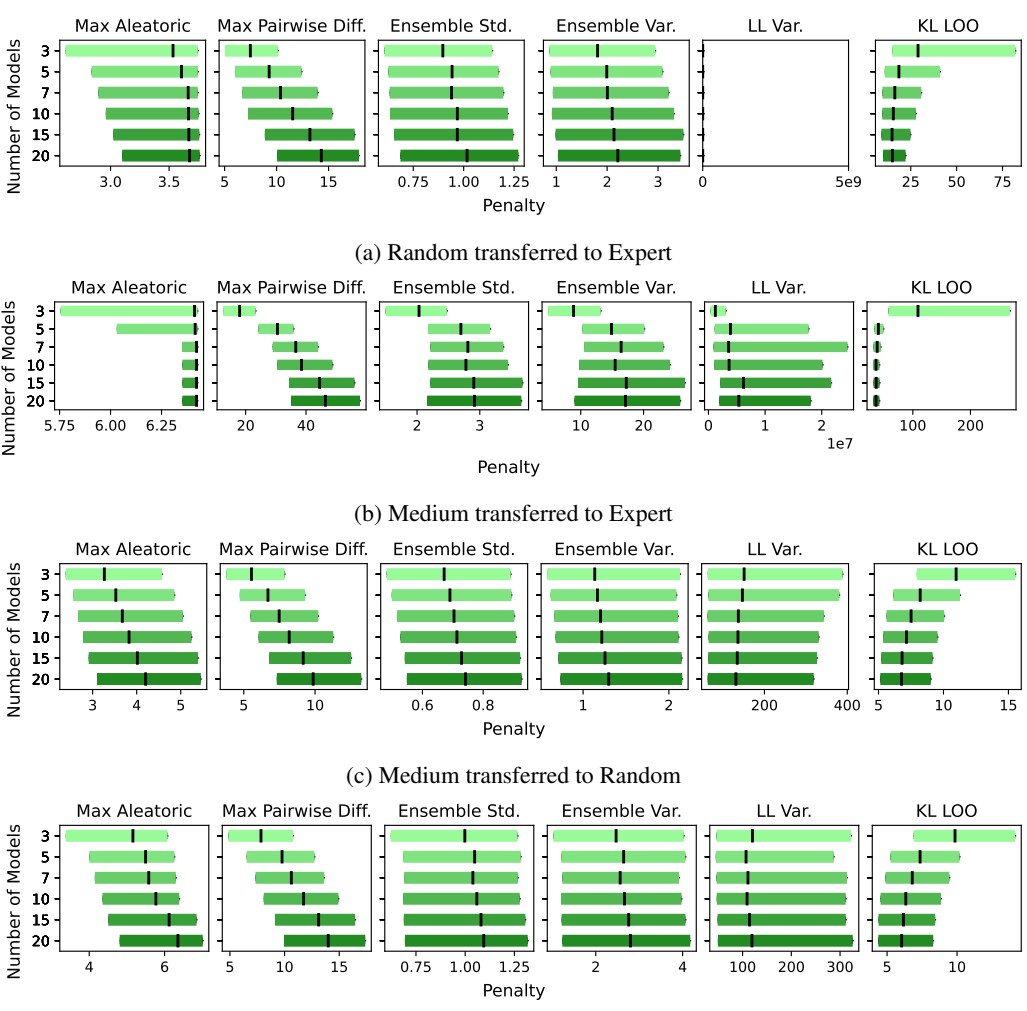

Figure 9: Box Plots showing HalfCheetah D4RL transfer tasks.

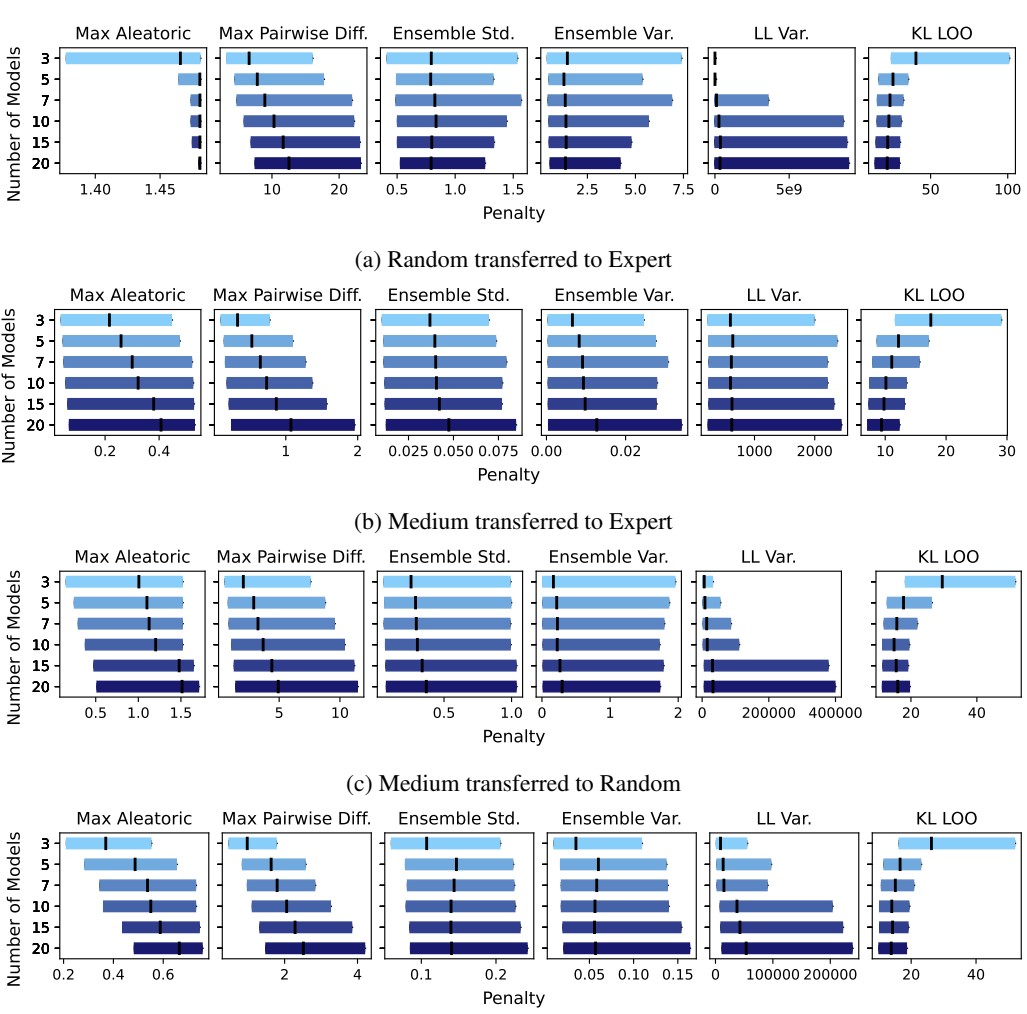

(a) Random transferred to Expert

(b) Medium transferred to Expert

(c) Medium transferred to Random

(d) Expert transferred to Random

Figure 10: Box Plots showing Hopper D4RL transfer tasks.

### B.1.2 TRUE MODEL-BASED ERROR DISTRIBUTION

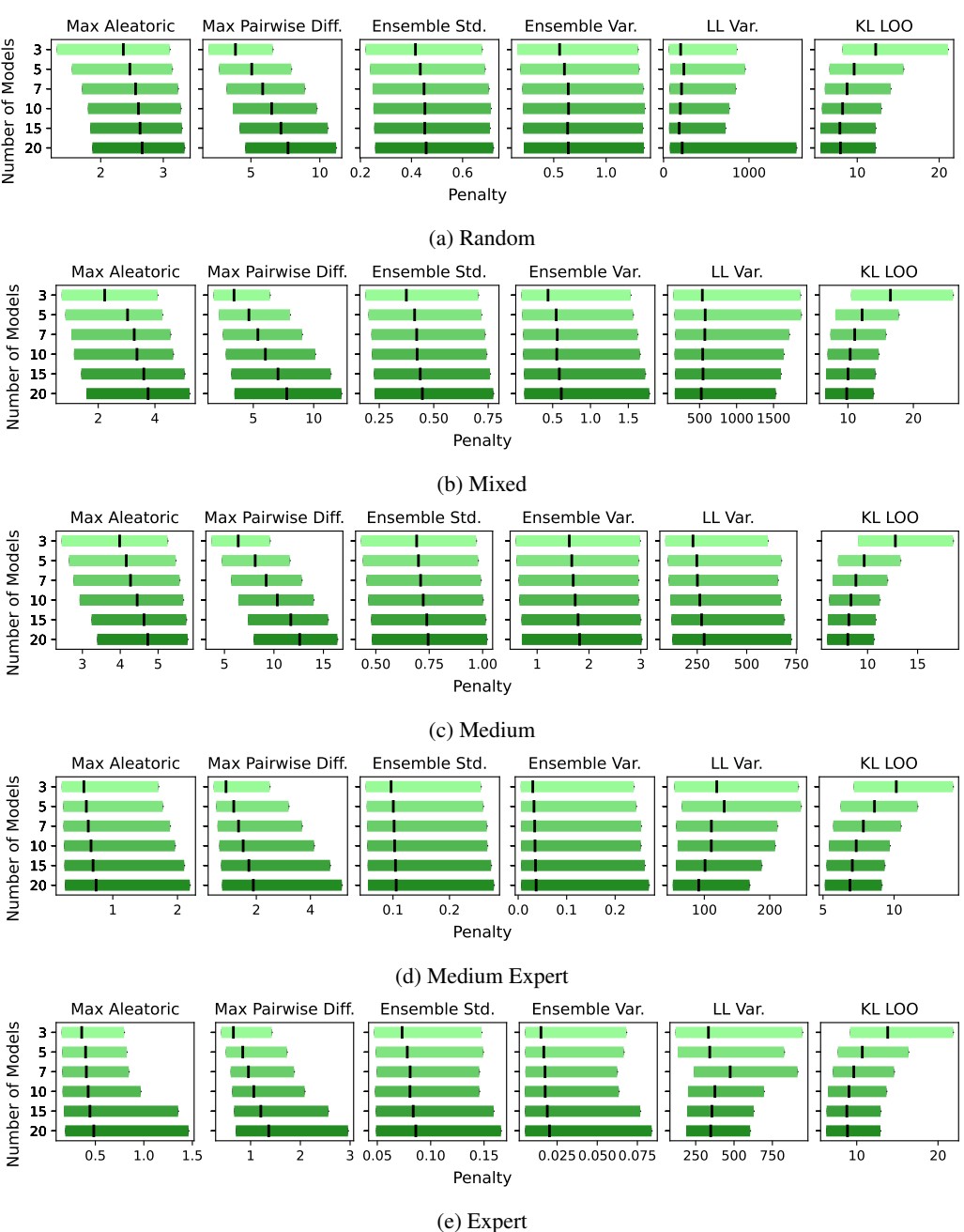

Figure 11: Boxplots showing HalfCheetah D4RL true model-based error penalty distributions.

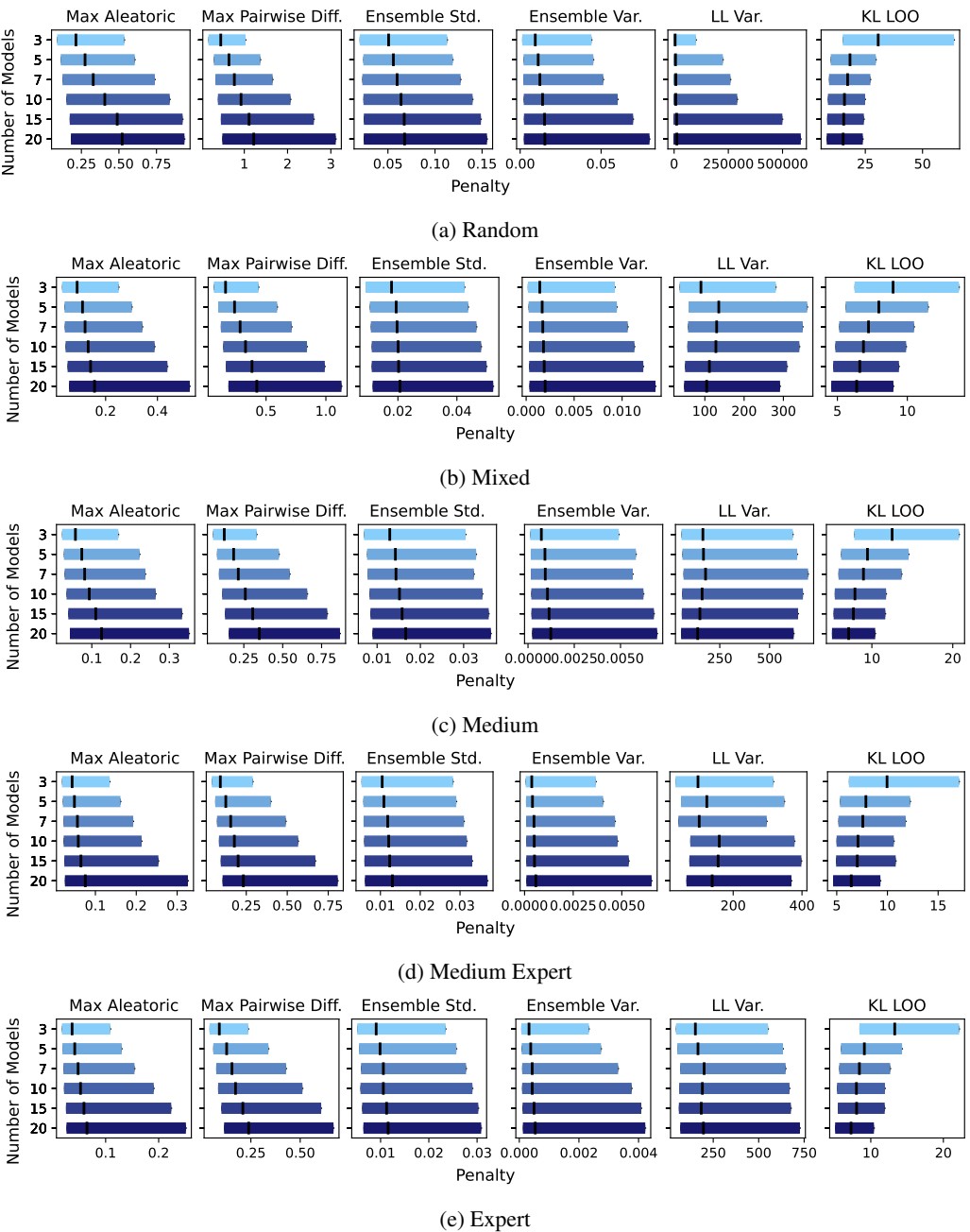

Figure 12: Boxplots showing Hopper D4RL true model-based error penalty distributions.

## B.2 Penalty Performance

In this section, we provide the full set of results showing the impact of increasing model count on the correlation statistics of each penalty, as described in Sec. 5.1. We show the Spearman and Pearson correlation between penalty and ground truth MSE for all training datasets. First, we present the transfer performance of all training sets onto all offline datasets. Then, we present the results for all training datasets under the True Model-Based experiment under the adversarial policies.

### B.2.1 HalfCheetah D4RL: Transfer

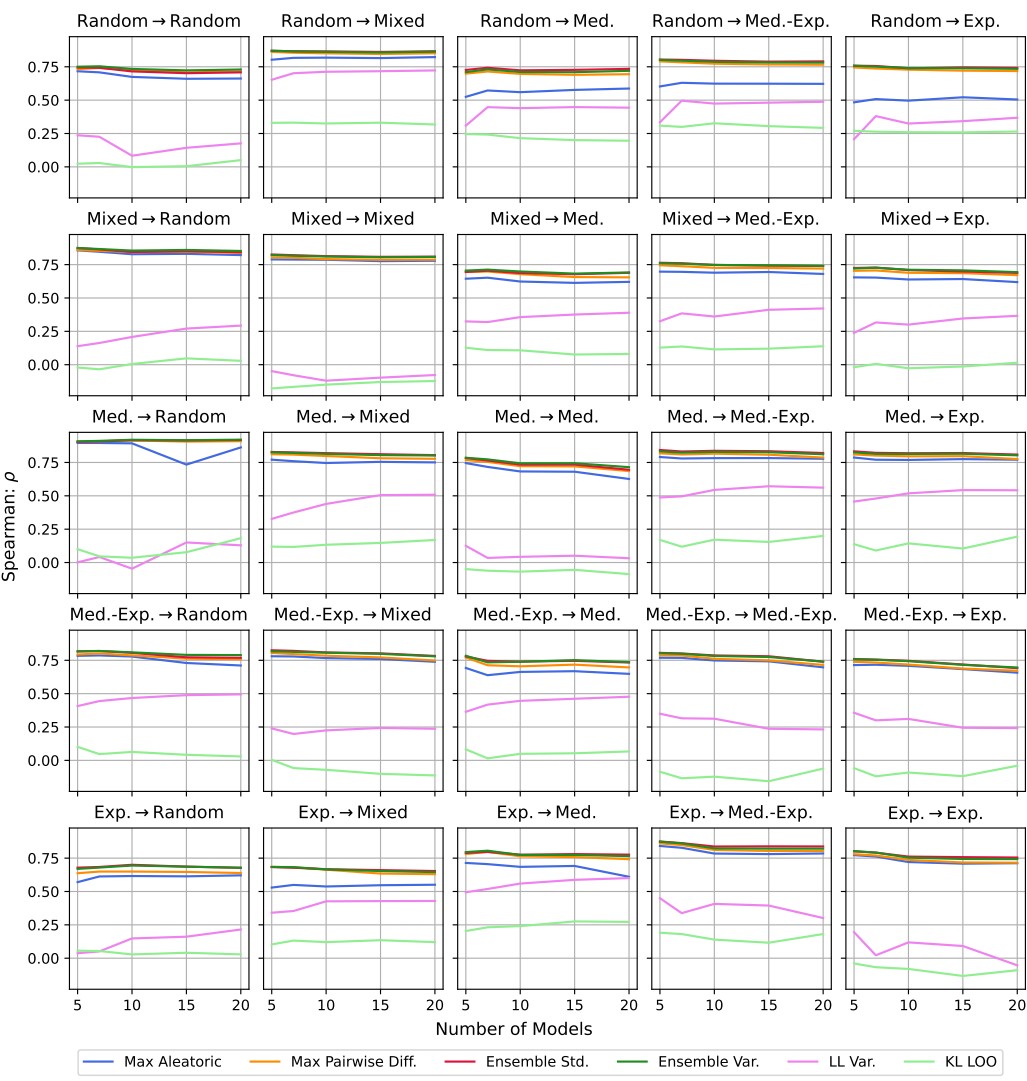

Figure 13: HalfCheetah Spearman Statistics

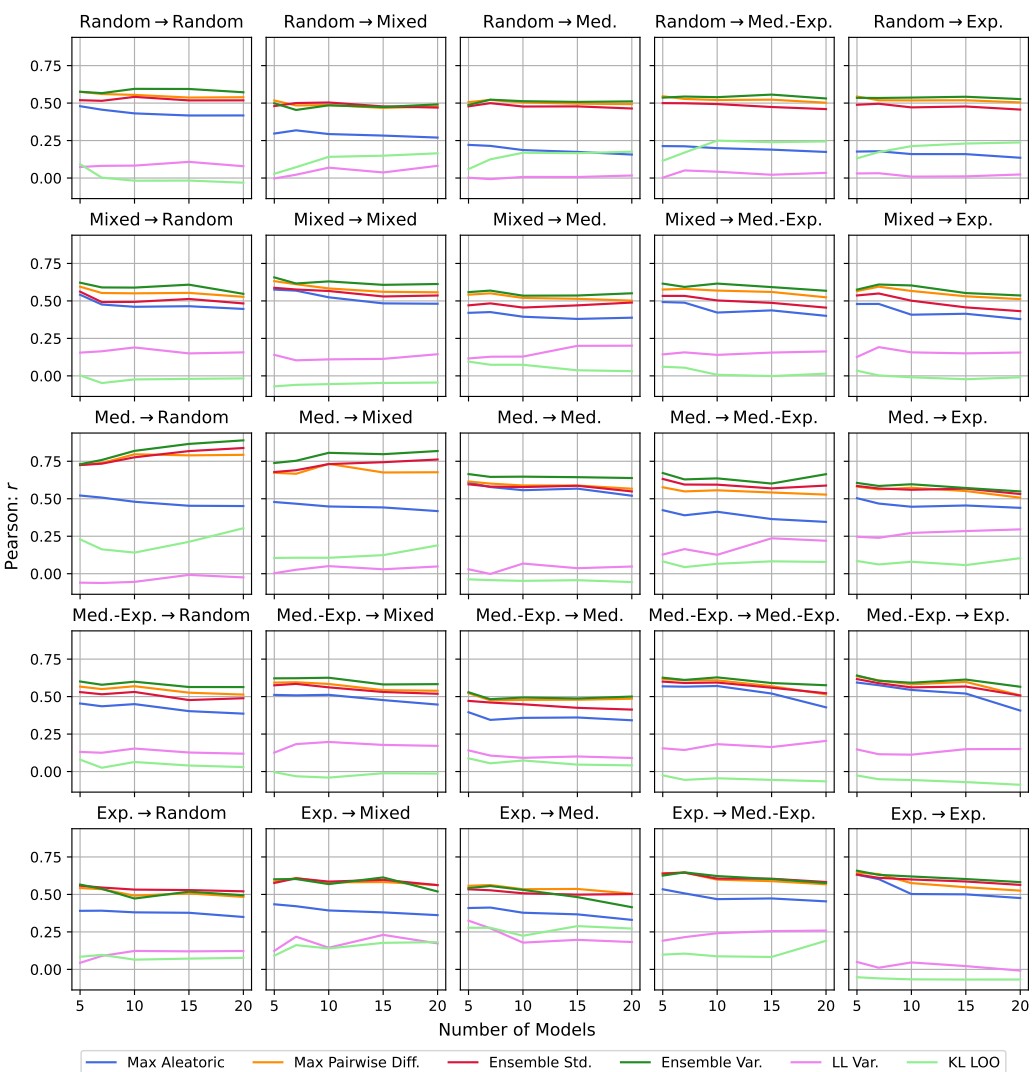

Figure 14: HalfCheetah Pearson Statistics

### B.2.2 Hopper D4RL: Transfer

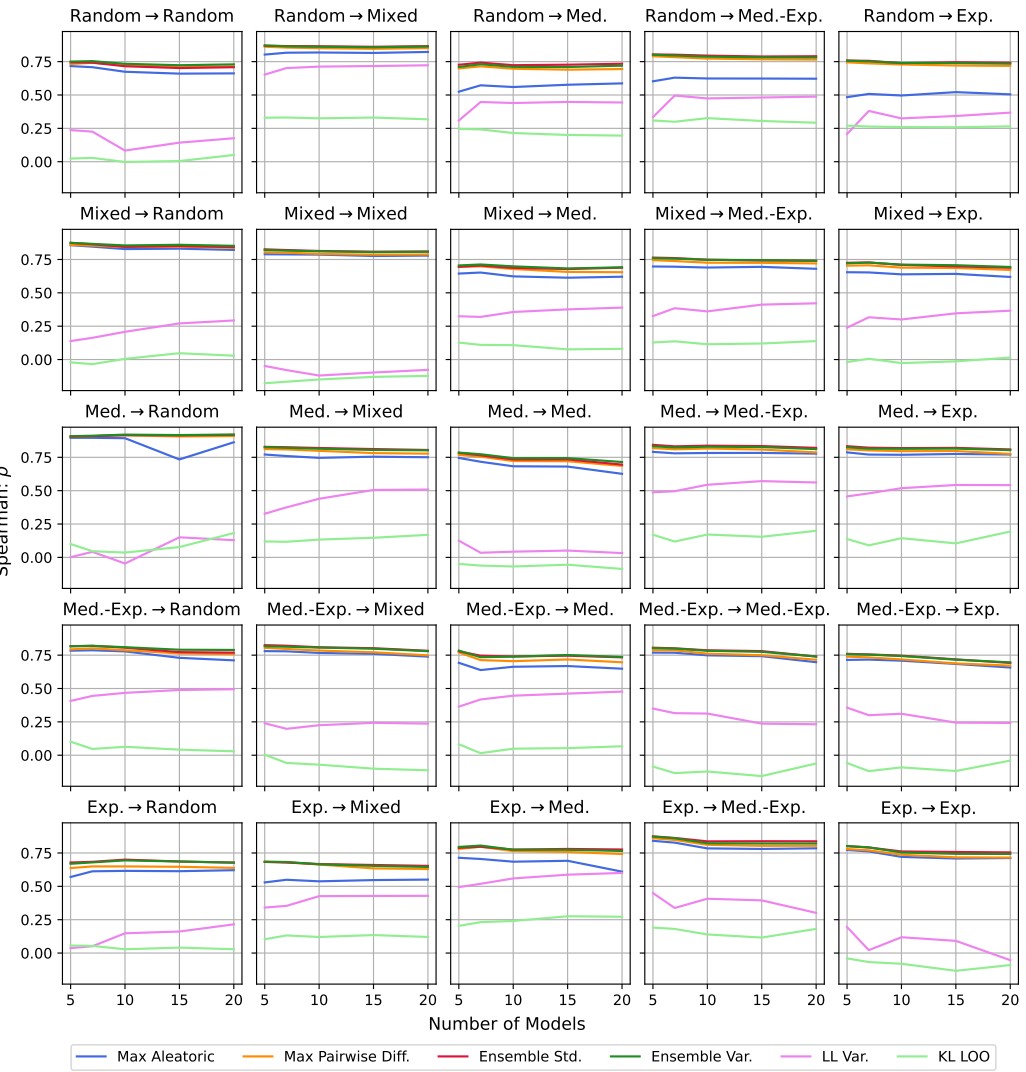

Figure 15: Hopper Spearman Statistics

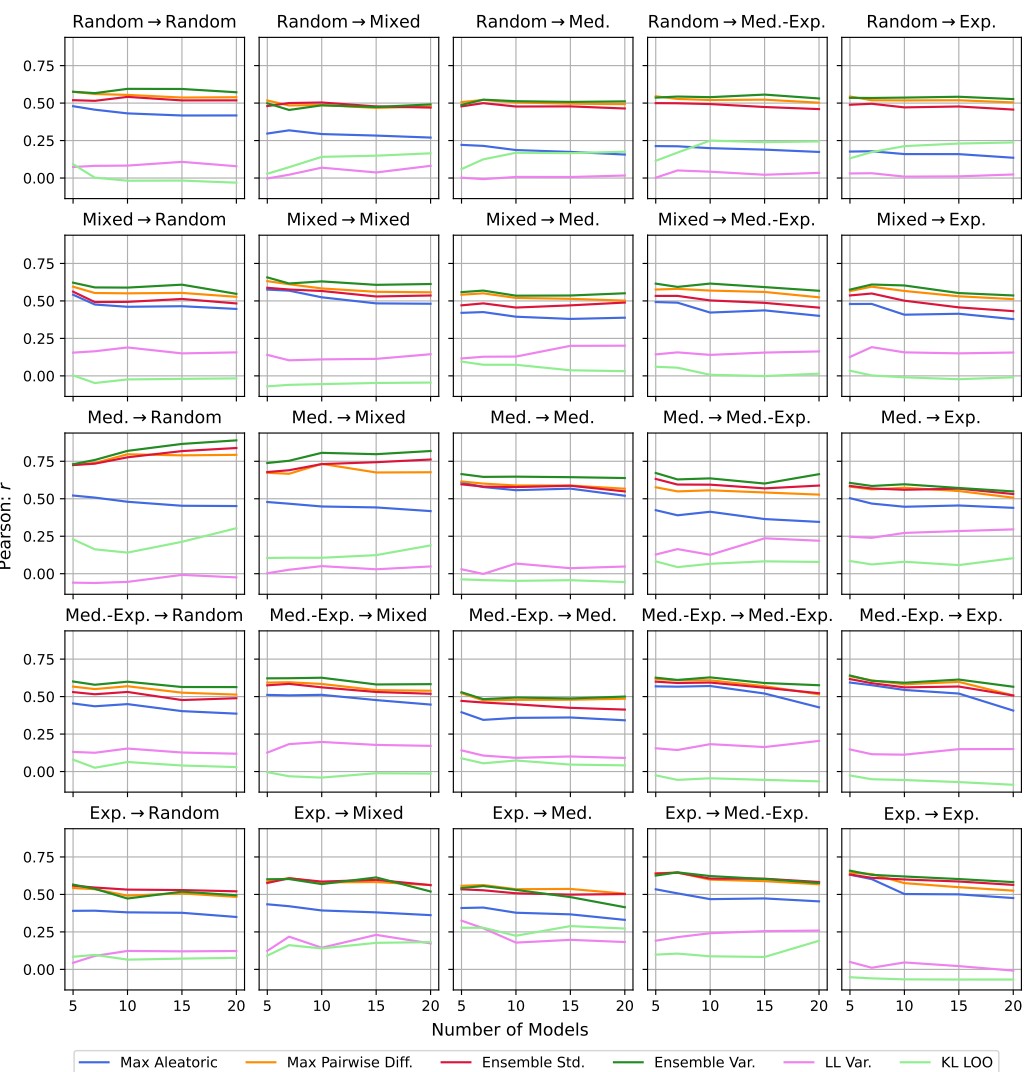

Figure 16: Hopper Pearson Statistics

### B.2.3  HALFCHEETAH D4RL: TRUE MODEL-BASED ERROR

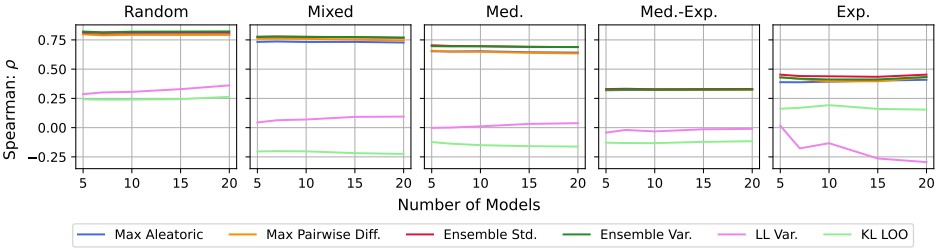

Figure 17: HalfCheetah Spearman Statistics

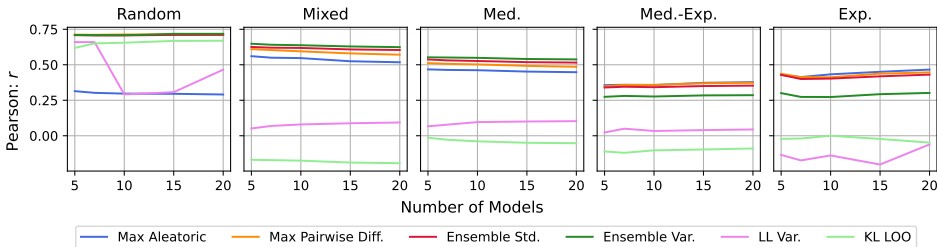

Figure 18: HalfCheetah Pearson Statistics

### B.2.4  HOPPER D4RL: TRUE MODEL-BASED ERROR

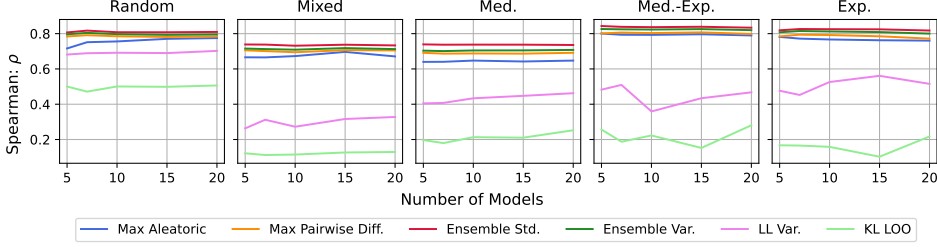

Figure 19: Hopper Spearman Statistics

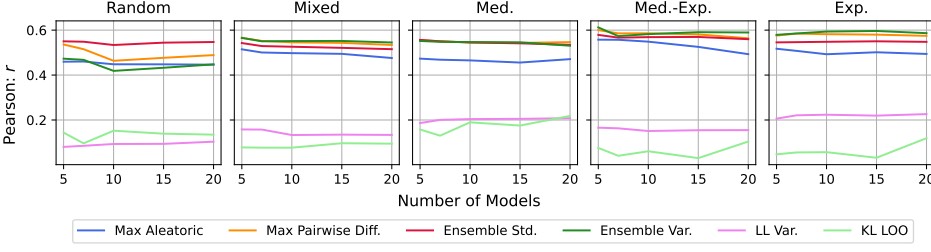

Figure 20: Hopper Pearson Statistics

### B.2.5 ALL AGGREGATED

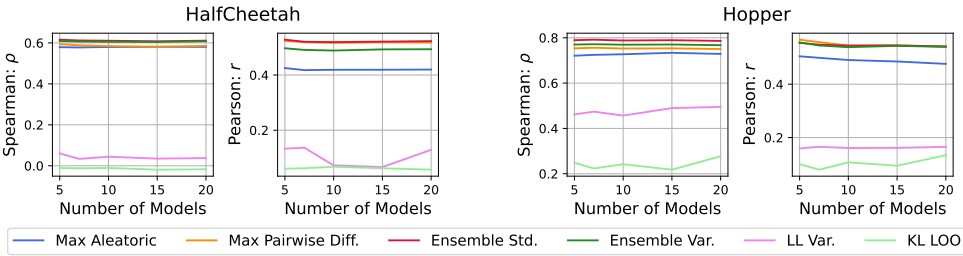

Figure 21: Aggregated True Model-Based correlation statistics over all datasets (i.e., Random through to Expert); **Left:** HalfCheetah; **Right:** Hopper

## C SKEWNESS AND KURTOSIS COMPARISONS

### C.1 SKEWNESS AND KURTOSIS OVERALL

Here we present the 3rd and 4th order statistics (skew and kurtosis respectively) of each penalty, illustrating that even with identical model counts, the shape statistics between penalties are vastly different.

Table 6: Skew ($\gamma_1$) and Kurtosis ($\gamma_2$) statistics of all experiments averaged over all datasets (i.e., Random through to Expert) using the MOPO Default of 7 models.

| | Transfer | | | | True Model-Based | | | |
| | HalfCheetah | | Hopper | | HalfCheetah | | Hopper | |
| Penalty | $\gamma_1$ | $\gamma_2$ | $\gamma_1$ | $\gamma_2$ | $\gamma_1$ | $\gamma_2$ | $\gamma_1$ | $\gamma_2$ |
|---|---|---|---|---|---|---|---|---|
| Max Aleatoric | -0.010 | 0.580 | 0.689 | 1.377 | 0.671 | 0.920 | 1.873 | 2.864 |
| Max Pairwise Diff. | 0.919 | 0.957 | 1.967 | 4.578 | 1.661 | 3.081 | 2.571 | 7.465 |
| Ensemble Std. | 0.794 | 0.806 | 2.136 | 6.560 | 1.656 | 3.178 | 2.739 | 9.061 |
| Ensemble Var. | 1.823 | 4.830 | 3.436 | 15.983 | 2.612 | 8.800 | 4.517 | 25.380 |
| LL Var. | 6.893 | 114.843 | 10.920 | 180.716 | 5.100 | 37.865 | 14.415 | 251.705 |
| LOO KL | 1.778 | 5.729 | 3.729 | 29.606 | 1.840 | 4.600 | 4.008 | 28.089 |

### C.2 SKEW AND KURTOSIS SCALING WITH MODEL COUNT

We omit LL Var. and LOO KL due to the fact that their changes were so significant as to obfuscate the changes of the more performant penalties.

We choose 7 models, as in Table 6, to act as our 'baseline' (following the default MOPO setting), and we measure the change in the skew and kurtosis relative to this, hence 7 models always has a 0% change in our graphs. For brevity, in the transfer experiments, we average over all 'transferred to' environments, e.g., Random, Medium, etc.; the graph title refers to the data that the model was trained on.

Again, we observe the environment *and* setting dependency of these metrics, sometimes having increasing skewness and kurtosis with model count, and other times decreasing. This further justifies using a ranking metric to compare penalties, as the overall penalty shape can vary hugely and unpredictably w.r.t. co-dependent hyperparameters. We do observe however in the True Model-Based experiments that ensemble standard deviation appears to be most robust to scaling with models. We also observe that the Max Aleatoric penalty can change shape significantly w.r.t. model count, and no penalties are fully immune to this. This further advocates the use of shape meta-parameters to control for changing distribution properties when adjusting the number of models as a hyperparameter, as well as selecting penalties that are relatively invariant to model count to make tuning easier.

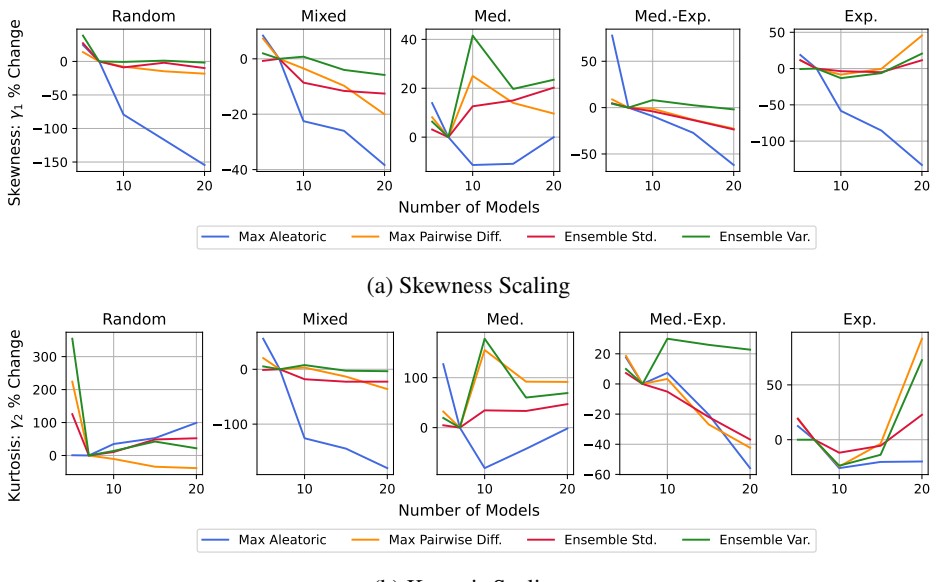

(a) Skewness Scaling

(b) Kurtosis Scaling

Figure 22: HalfCheetah Transfer.

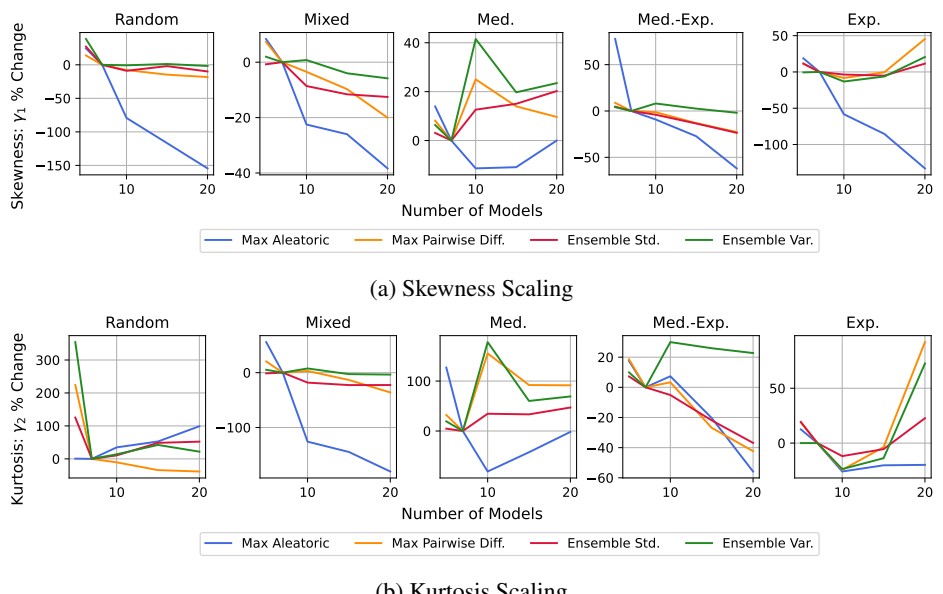

(a) Skewness Scaling

(b) Kurtosis Scaling

Figure 23: Hopper Transfer.

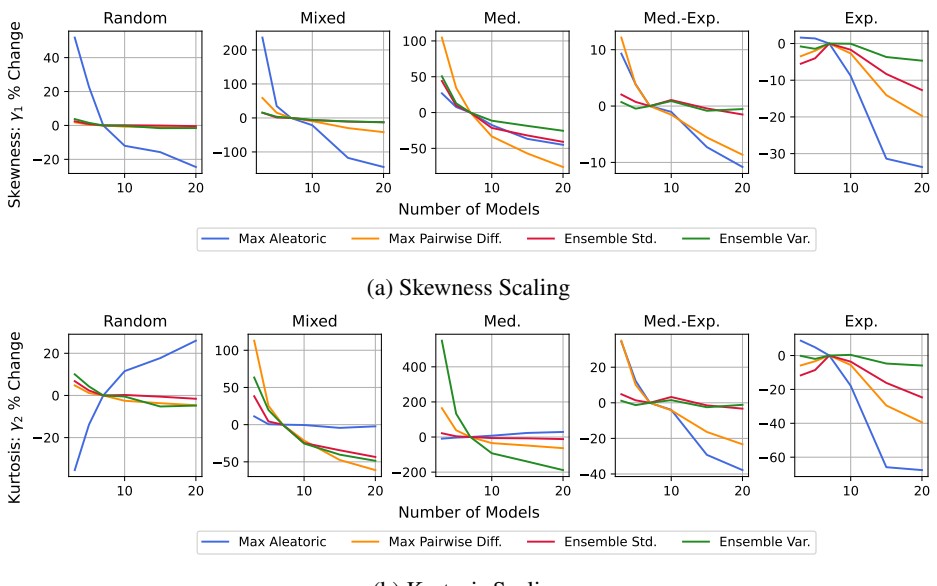

(a) Skewness Scaling

(b) Kurtosis Scaling

Figure 24: HalfCheetah True Model-Based.

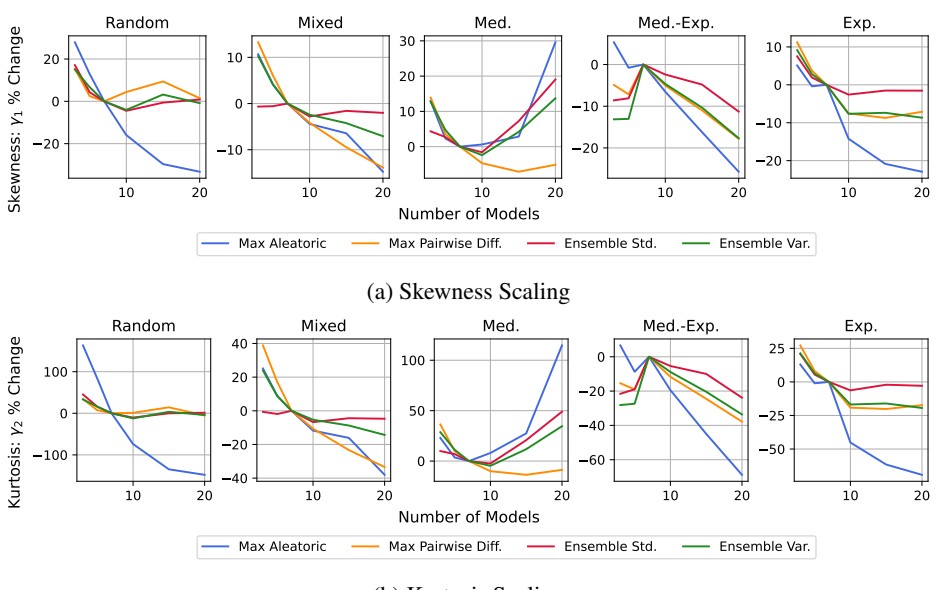

(a) Skewness Scaling

(b) Kurtosis Scaling

Figure 25: Hopper True Model-Based.

## D    FURTHER DETAILS ON TRUE MODEL-BASED EXPERIMENTS

### D.1    METHODOLOGICAL DETAILS

We leverage the MuJoCo (Todorov et al., 2012) simulator to provide us with ground truth dynamics that we can use to compare against our model predictions and penalties. This is done by providing the state and action inputs given to the model to the simulator through the `set_state` method in the simulator API. It must be noted that this method also requires an addition 'displacement' value which is not modelled by the world models (nor is it provided in the D4RL data), however we found in practice this did not affect the dynamics predicted by the simulator, and simply setting this to $0$ was sufficient to generate ground truth predictions.

This makes it possible to provide the simulator the hallucinated model states, and provide a true proxy to the *dynamics* discrepancy. We note that since the states are 'hallucinated' by the model, it might be the case that they may not be admissible under the true environment, but in reality the simulator was able to process almost any combination of state and action, barring settings that featured anomalously large magnitudes. To handle such cases, we found it necessary to clip the model states to the range $[-10, 10]$.

In order to assess the permissibility of states, as well as measure the accuracy of the penalties as OOD input detectors, we provide an alternative distance measure based on the distance away from the training set. We use this measure for our analysis in Section 5.3, and is calculated as the distance from the offline training dataset, which we define to be the 2-norm between a given state-action tuple and its nearest point in the offline data, a similar metric to those used in recent works on imitation learning (Dadashi et al., 2021). We describe this quantity henceforth as 'Distribution Error'.

### D.2    ON THE NATURE OF OOD DATA ALONG HALLUCINATED TRAJECTORIES

Here we discuss the nature of OOD data along a single hallucinated trajectory (in the model) in offline MBRL, analyzing the inductive bias that some 'error' increases with increasing rollout length in the model. We find that there is merit to this assumption, and show this in Fig. 26 for all HalfCheetah and Hopper environments in D4RL. Here, we plot the median error at each time-step across $30,000$ aggregated trajectories in the model. Note that for all plots, we re-normalize all penalties by subtracting their mean and dividing by their standard deviation to facilitate comparison; this normalization was also applied in the analysis performed in Fig. 1a. Concretely, each time step corresponds to the normalized median value of $30,000$ data-points.

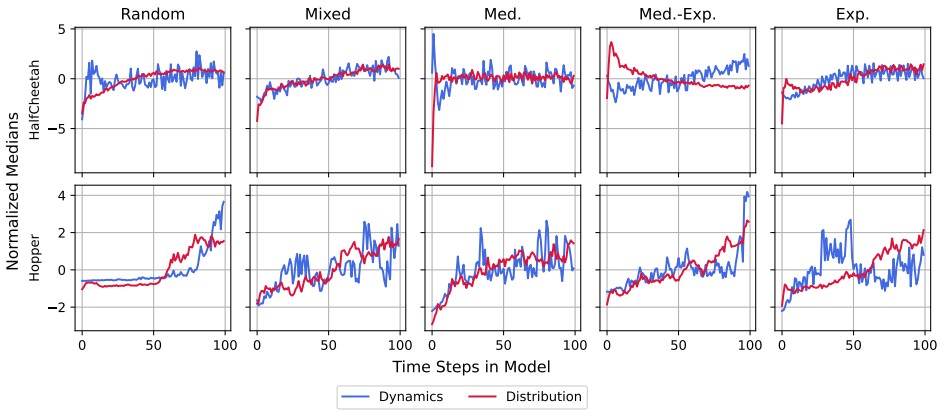

Figure 26: Median True Model-Based Errors as a function of rollout timestep

We observe indeed that both median dynamics and distribution errors increase with increasing time step in the model. The only real exception is HalfCheetah Medium-Expert, which we believe to be due to our trained policy not being able to successfully exploit this environment.

The above analysis captures *overall trends* in the error over a large number of trajectories. However, the way errors manifest during an *individual rollout* is not so straightforward. To illustrate this,

observe Fig. 27, where we plot a random subset of 5 individual rollouts from the Hopper Medium-Expert data we generated.

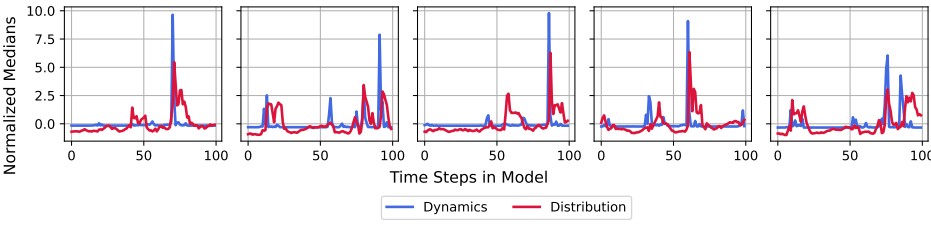

Figure 27: Several Individual Ground Truth Rollouts in Hopper Medium-Expert

We observe that errors along any single trajectory tend to manifest as 'spikes', and that it is entirely possible to recover from these, returning to either admissible dynamics, or parts of the state-action space that have been seen in the data. This speaks to the nature of how we ought to penalize policies for accessing regions of inaccuracy/uncertainty, and may justify a hybrid MOPO/MOReL approach, whereby we penalize individual transitions along a trajectory, but do not stop rollouts early. Indeed, this is similar to the approach taken in M2AC (non-stop), albeit they choose to 'mask' uncertain transitions, not penalize them. We leave the design of such an algorithm to future work.

Finally, we address the issue of comparing OOD dynamics and inputs. As already observed in Fig. 27, these two errors are not necessarily always the same, and oftentimes it is possible that one quantity is large, whilst the other is small. We revisit Fig. 1a to explore this, now also plotting the Distribution Error in Fig. 28.

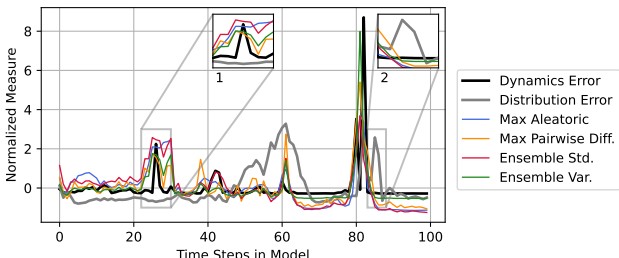

Figure 28: Comparing OOD dynamics and inputs on a Hopper Medium-Expert trajectory

We first speak to the inset annotated '1'. Here we observe that the transitions generated in fact closely resemble the data that our model was trained on, however the predicted dynamics are incorrect, and cause an aforementioned 'spike'. This is the opposite of what is observed in the inset annotated '2'; where we actually predict accurate dynamics, however the resultant state-action tuples do not closely resemble the data that our model was trained on. We generally observe that regions of high Distribution Error tend to be preceded by 'spikes' pertaining to high Dynamics Error, and this present an exciting avenue for future work understanding how these quantities are related.

### D.3 ON THE DIVERSITY OF EXPLOITATIVE POLICIES

It is possible that training policies purely to exploit the world models may result in generating state-action tuples that are low in diversity, as the policy could discover "pockets" in the model that provide consistently high return. To prevent this, we train multiple policies inside the model from different seeds, with the aim of inducing different modes of exploitation. To validate this induces diverse trajectories in the world models, we visualize the state-action manifold using a t-SNE projection (van der Maaten & Hinton, 2008) of: 1) the D4RL Hopper Mixed-Replay (which contains diverse samples); 2) the imagined rollouts inside the model from the exploitative policies in Fig. 29. The policies were trained to exploit a model that itself was trained on the Hopper Mixed-Replay data.

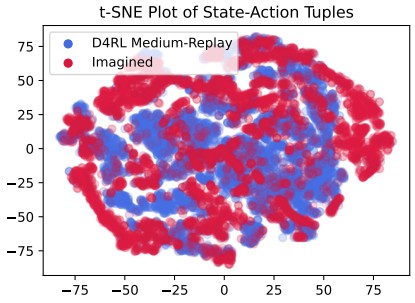

Figure 29: t-SNE projection of 10,000 Hopper D4RL Medium-Replay and 10,000 exploitative Imagined policy state-action tuples

We observe that the induced policies inside the model displays some overlap with the D4RL data, but also resides in parts of the manifold where there is little coverage from the D4RL data, likely representing regions of exploitation. Importantly, the exploitative WM trajectories display strong state-action tuple diversity, comparable to that of the offline data it was trained on.

# E    USING METRICS AS OOD EVENT DETECTORS

## E.1    MEASURING STATISTICS

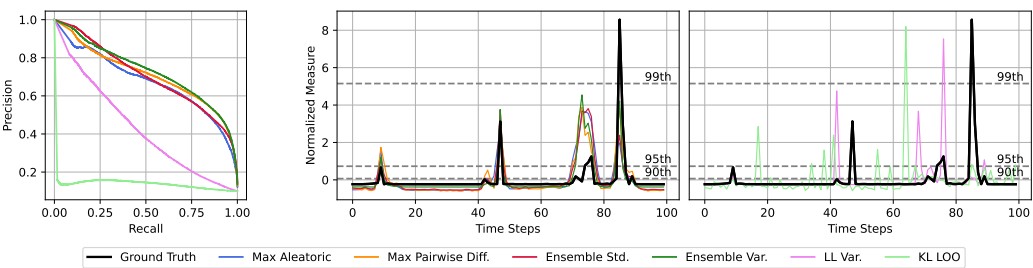

Figure 30: Hopper Medium-Expert True Model-Based Experiments; **Left:** Precision v.s. Recall against Ground Truth; **Middle:** Higher Performing Penalties v.s. Ground Truth MSE in Imagined Rollout; **Right:** Lower Performing Penalties v.s. Ground Truth MSE in Imagined Rollout

As noted previously, different penalties have varying scales and distribution profiles, so we need a way of standardizing the method of assessment. Using our observation that errors manifest as 'spikes' during a rollout, we propose treating each penalty as a classifier. Concretely, our test set consists of the ground truth data labeled by whether or not they exceed a certain percentile at a particular time step. Each penalty may be then be treated as a 'classifier' by normalizing its range to lie in $[0, 1]$. We can then use standard classification quality measures, such as AUC, to determine the effectiveness of these penalties at capturing these spikes, whilst sidestepping the issue of the different distributional profiles identified previously.

Fig. 30 shows how our proposed method may be used to compare the effectiveness of each metric at capturing OOD events. In the figure, we plot a single rollout in the model, and the resultant ground truth MSE between the predicted next state and the true next state in black. We then superimpose the 90th, 95th and 99th percentile MSEs across the entire imagined trajectories onto the figure in gray dashed lines. To construct our OOD labels, we label any point below the percentile line as being 'False', and any point above that line as being 'True'. Finally, we normalize the uncertainty metrics as previously described into values in the range $[0, 1]$, allowing us to construct precision-recall graphs and calculate classifier statistics.

## E.2 PRECISION RECALL CURVES

In this section we present the Precision-Recall curves described in App. E.1.

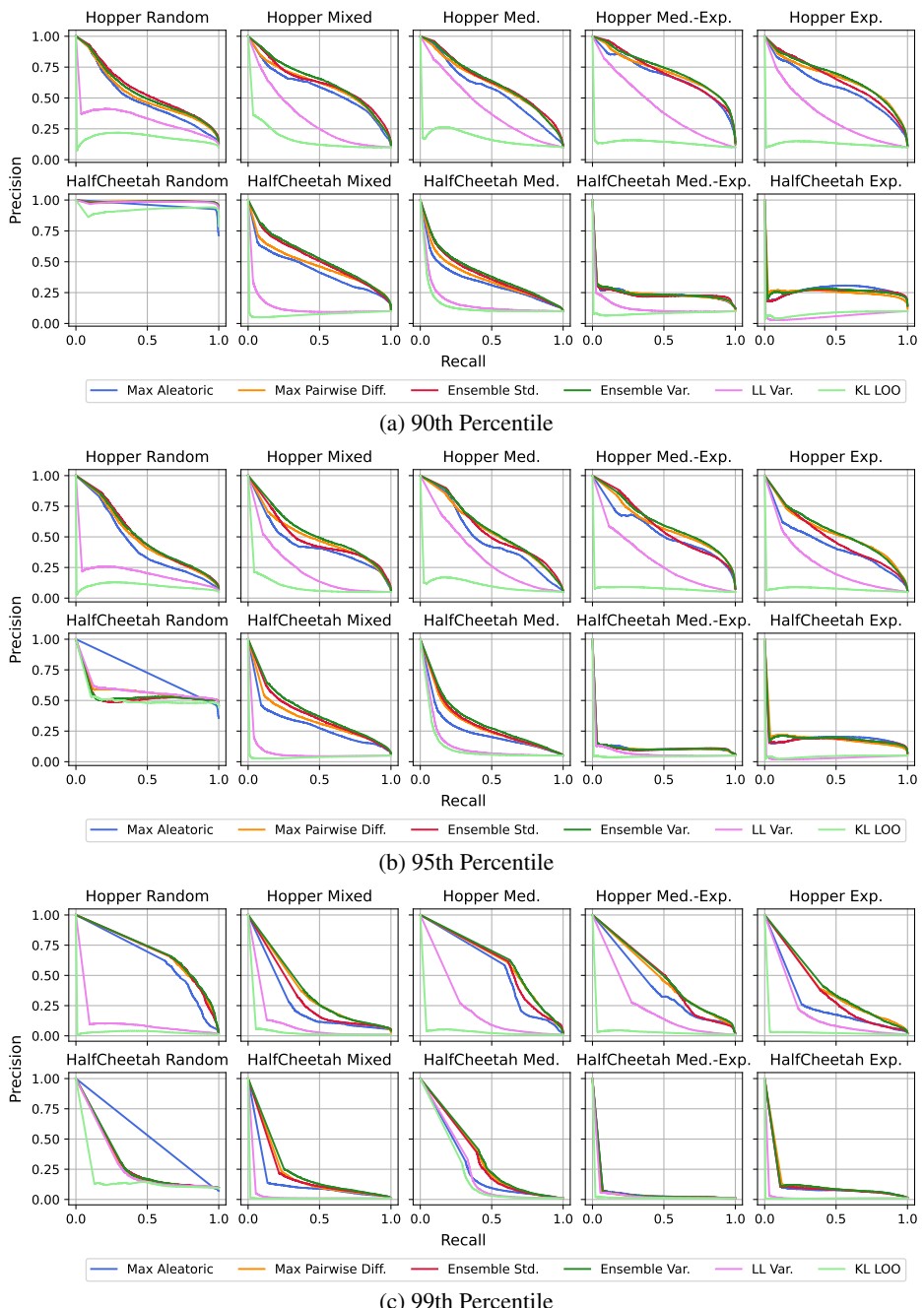

Figure 31: Precision Recall curves on ground truth data.

## F    IMPLEMENTATION DETAILS

This paper extensively discusses key *hyperparameters* specific to current offline MBRL algorithms. However, there are significant *code-level* implementation details which are often critical for strong performance and make it hard to disambiguate between algorithmic and implementation improvements. Worryingly, many of these details are not mentioned in their respective papers, or are different between the authors' code and paper. We detail clear examples of this below. We believe further investigation of these code-level implementation details represents important future work, as has already been done for policy gradients (Engstrom et al., 2020; Andrychowicz et al., 2021). Indeed – it is unclear if the improvement of MOReL over MOPO is due to its different P-MDP formulation, or if it is successful *in spite of* this formulation, due to a superior policy optimization strategy or dynamics model design. We believe that this paper takes a significant first step in tackling this issue by directly comparing a number of key design choices, and understanding their individual impact. Now we summarize key differences between the paper and code for the MOPO and MOReL algorithms which we compare against that are crucial to achieve the same reported performance.

In MOPO,

- Each layer in the model neural network has a different level of weight-decay

- The authors' code uses different objectives for training (log-likelihood) and validation (MSE).

- The authors use elites, but only for next state prediction (as discussed previously).

In MOReL,

- There is a difference in the authors' code about how the penalty threshold is calculated and tuned, and isn't provided as a hyperparameter in the appendix.

- The absorbing HALT state does not appear in the authors' code.

- The negative halt penalty appears significantly different between code and paper.

- There is a minimum trajectory steps parameter (hard-coded to 4) not mentioned in the paper.

- The reward function appears to be hard-coded in the authors' implementation, not learned as stated in the paper.

- The policy architecture is different in the authors' code (64,64 hidden layers) and the paper (32,32 hidden layers)

- It is not clear when the optional behavior cloning initialization step is applied.

## G    HYPERPARAMETERS AND EXPERIMENT DETAILS

The D4RL (Fu et al., 2021a) codebase and datasets used for the empirical evaluation is available under the CC BY 4.0 Licence. As stated in the main text, we choose to use 'v0' experiments as these are more challenging for Hopper due to having low return trajectories (Kostrikov et al., 2021), and we clearly state when other benchmarks use the 'v2' experiments, which have offline trajectories with higher returns on Hopper.

The remaining hyperparameters for the MOPO algorithm that we do not vary by Bayesian Optimization were taken from the original MOPO paper (Yu et al., 2020), apart from we fix the number of policy epochs/iterations to 1,000 for all experiments. This means our implementation uses the same probabilistic dynamics models (with unchanged hyperparameters) and policy optimizer (SAC, Haarnoja et al. (2018)) as MOPO, differing from MOReL, which uses Natural Policy Gradient (Kakade, 2002).

The hyperparameters used for the BO algorithm, CASMOPOLITAN, are listed in Table 7. We use the batch-mode of CASMOPOLITAN, where multiple hyperparameters settings are proposed and evaluated concurrently.

Each BO iteration is run for 300 epochs on a single seed, and the full optimization over an offline dataset took ~200 hours on a NVIDIA GeForce GTX 1080 Ti GPU taken up predominantly by MOPO training.

Table 7: CASMOPOLITAN Hyperparameters

| Parameter | Value |
|---|---|
| Number of parallel trials | 4 |
| Number of random initializing points | 20 |
| ARD | False |
| Acquisition Function | Thompson Sampling |
| Global BO | True |
| Kernel | CoCaBo Kernel (Ru et al., 2020) |

Unless specified otherwise, plots and reported statistics are completed with 7 models in the ensemble, as this is the number chosen in the original MOPO paper used with the Max Aleatoric penalty.

## H   RLIABLE FRAMEWORK FOR PERFORMANCE EVALUATION

Throughout this work, we choose to adopt the `rliable` framework introduced in Agarwal et al. (2021) to evaluate the performance of our approaches. `rliable` advocates for computing aggregate performance statistics and probability of improvement across many tasks in a benchmark suite; indeed we take this approach when reporting the values in the analysis performed in Sections 4 and 5. This is important when the number of tasks become large, and also prevents outliers from dominating mean statistics. Furthermore, this allows us to make clear statements about improvements given the relatively low number of seeds that are used in deep RL; indeed, Agarwal et al. (2021) show that even using high seed counts does not ameliorate the variance issues experienced when training such algorithms. For normalization, we use the standard D4RL return scaling.

## I   EVALUATION USING AUTOMATIC CONSTRAINT TUNING

We show the full tabulated results from Sec. 6 with statistical significance using the `rliable` framework in Table 8, using the 'Probability of Improvement' metric in Agarwal et al. (2021).

To evaluate our claims in Sections 4 and 5 without needing to laboriously tune the penalty weight $\lambda$ per environment, we employ an automatic penalty tuning scheme, analogous to the automatic entropy tuning used in Haarnoja et al. (2018). Concretely, given a constraint value $\Lambda$, at each epoch we minimize:

$$J(\lambda) = \mathbb{E}_{\mathbf{s}_t, \mathbf{a}_t \sim \mathcal{D}} \left[ \log \lambda (\Lambda - \lambda \cdot u(\mathbf{s}_t, \mathbf{a}_t)) \right] \tag{5}$$

We start from an initial weight $\lambda = 1$. We observe that the penalty weight found by automatic tuning tends to converge within the first 50 epochs and then remains stable throughout training.

Table 8: Improvement over grid-searched MOPO through restricted hyperparameter choices (e.g., one single choice, or an $\arg \max$ between two) on the D4RL MuJoCo benchmark. The single and two setup approaches both use the Ensemble Std. penalty and $N = 10$.

| Algorithm | Average Score | $\mathbb{P}$[Improvement over MOPO] |
|---|---|---|
| MOPO (default hyperparameters) | 34.2 | - |
| Single setup: ($h = 20$, $\Lambda = 1$) | 49.0 (+43%) | 73.96% |
| Two setups: $\arg \max \{(h = 10, \Lambda = 0.5), (h = 20, \Lambda = 1)\}$ | 57.8 (+69%) | 80.20% |
| Optimized MOPO (ours, Table 3) | 65.2 (+91%) | 89.06% |

## J  BEST FOUND ADROIT HYPERPARAMETERS

We present the best found hyperparameters under our BO procedure for the D4RL Adroit tasks in Table 4. We see similar trends as in our main evaluation in Table 3 favoring higher rollout lengths and the Ensemble penalties.

Table 9: Best discovered hyperparameters using BO for Adroit

| Environment | | Discovered Hyperparameters | | | |
|---|---|---|---|---|---|
| | | N | $\lambda$ | h | Penalty |
| pen | cloned | 10 | 6.64 | 12 | Ensemble Std |
| | human | 11 | 0.96 | 37 | Ensemble Var |
| | expert | 7 | 4.56 | 5 | Max Aleatoric |
| hammer | cloned | 10 | 0.21 | 12 | Ensemble Var |
| | human | 13 | 2.48 | 47 | Ensemble Std |
| | expert | 12 | 0.99 | 37 | Ensemble Std |

