# OpenReview forum: "Revisiting Design Choices in Offline Model Based Reinforcement Learning"
_ICLR.cc/2022/Conference — ICLR 2022 Spotlight_

### Official Review · Reviewer_PM7Z · 2021-10-30

**Correctness:** 4
**Technical Novelty And Significance:** 3
**Empirical Novelty And Significance:** 3
**Recommendation:** 8
**Confidence:** 4

**Main Review:**

Strengths
- the paper is well written and overall easy to read and understand.
- such work, summarizing different contributions and comparing the different choices that are made in each contribution, are important for the research community as well as practitioners
- I also like the different messages and perspectives that are made in the paper
- The related work seems rather complete for model-based offline RL.

Weaknesses
- I would need a better explanation of Figure 1a and section 5.3. (see detailed comments below).
- for such an empirical work error bars are needed in Table 1, 2, 3, 4 to judge the statistical significance of the results. Table 3 identifies algorithms that are significantly better but it would still be nice to have corresponding std or error bars for the different performances.
- What about existing calibration metrids such as the expected calibration error (ECE)?

Other comments:
- Figure 1:
    * the figure should be better documented, even if this has to be in the appendix. A precise definition of the y-axis of FIgure 1a should be given and a precise description of how this figure was computed so that someone can reproduce it. What's the environment? Is the error and uncertainty the ones for the reward or another dynamics variable (or the mean error over all the dynamics state variables)
    * I would especially like the authors to clarify whether the errors are computed step by step comparing the outputs f(s,a) (obtained with the true dynamics) and g(s,a) (obtained with the model) assuming same inputs (s,a) or for multi-step predictions i.e. considering a fixed sequence of actions and a first initial state s_0 look at the error between each state of the sequence (s_1, ..., s_t) predicted by a rollout with the model and its corresponding state in the sequence (s'_1, ..., s'_t) obtained for the same action sequence with the true dynamics when starting from the same initial state s_0. Is it what is being computed in Figure 1a and in Figure 27? What is the difference between figure 26 and figure 27: Figure 26 is just the median overall several rollouts whereas figure 26 shows individual rollout?
    * I think this is a bit more clear in appendix D, but I would still like the authors to make this clear in their replies and to make the effort of making this more clear in the main paper as well.
    * Appendix D: Can the authors elaborate on the "displacement" variable? Is this a variable needed when setting the state of the Mujoco Simulator?
    * for Figure 1b: the authors should clarify the definition of the probability of improvement and how this is computed
- For figure 1 and table 1: what if the error used is the log likelihood instead of the MSE?
- Can the authors confirm that the hyperparameter optimization is done by evaluating the performance of the different hyperparameters on the real environment? My comment here is that this is impossible in practice. I would not let this influence my review because I think this is the current practice for most papers in RL.
- Table 2: "the penalties are powerful at identifying dynamics discrepancy but not as accurate at identifying when the world-model data is out-of-distribution" -> it seems that AUCs of 0.9 are reachable which is not too bad in my opinion?

Minor Comments
- Figure 1a legend: during single a model rollout -> during a single model rollout?
- Introduction: "Furthermore, using the insights gained from this section" -> it is not clear at all what "this section" is refering to here.
- Section 3. reward notation "r" is also used for the Pearson bivariate correlation.
- equation (1) -> I think a capital "R" should be used for the reward function.
- I should maybe have a look at the paper by Yu et al, 2020 but how do they justify the use the aleatoric uncertainty as a penalty? Aleatoric uncertainty is specific to the system and even the true dynamics have an aleatoric uncertainty. I would expect better results by considering epistemic uncertainty which is what is attempted when using the ensembles.

**Summary Of The Paper:**

Recent successful model-based offline RL techniques have relied on heuristics to penalize rewards according to the uncertainty of the estimated MDP. This paper reviews the different penalties that have been designed in the literature. The impact and importance of associated hyperparameters such as the planning horizon and the number of models in the ensemble are also evaluated. The author show that the selection of the best penalty and best hyperparameters lead to stronger performance.

**Summary Of The Review:**

I like the paper and the work done by the authors. I think it is valuable for the community (researchers and practitioners). I would just like to have some clarifications about the points I raised in my review. I will change my score accordingly.

---

> ### Author Response · Authors · 2021-11-18
> **Response to reviewer PM7Z (Part 1)**
>
> Thank you for your detailed and thoughtful review, we are very pleased to see you found our work well-written and could see the value of our work to the wider RL community. We hope this response addresses the concerns and questions raised in your review. Please let us know if any other issues remain.
>
> **“Error bars on tables”:**
> Thank you for pointing this out, we have updated the manuscript with error bars across seeds.
>
> **“ECE instead of MSE?”:**
> We choose MSE as the environment is deterministic, and furthermore whilst ECE is well-defined for classification, it is less so for regression (although there is some preliminary work designed to create analogues). We also note that MSE is motivated by theory, whereby assuming a Wasserstein form of the IPM (instead of the total variation distance) in Eq 1, we retrieve a $|G|$ that represents the MSE between the true and approximate dynamics. Through this lens, the MSE can be considered the ‘oracle’ from which we can derive an upper bound over the expected return, hence why being strongly correlated to this metric is vital for good performance as it enforces a tighter lower bound on expected return. We demonstrate this empirically, as more strongly correlated metrics get chosen as being optimal under Bayesian optimization.
>
> **“Better explain Fig 1a”:**
> Thank you for pointing this out. To clarify: the offline environment is Hopper Medium-Expert. The error measured is MSE dynamics error. The y-axis was normalized by rescaling all metrics by their mean and standard deviation; we will include this detail in Appendix D when discussing this general methodology.
>
> **“Clarification of rollout error calculations”:**
> In these experiments, we produce all trajectories inside the model (selecting actions under the policy), and compare each next state prediction with the true environment (simulator) prediction. This means s_0,...,s_T are all derived from the model, not the simulator. What we validate is whether the predicted *next state* at each timestep is correct under the simulator (which we do by resetting the state of the simulator to the states predicted under the world model at each time step). This allows us to measure the calibration of uncertainty under the actual covariate shift induced by rolling out *in the model*. This covariate shift arises due to a) a different behavior policy than the offline data; b) trajectories being generated from the world model autoregressively, not being gathered from the ‘true environment’ (which is the source of the offline data we trained our world models from). In other words, these experiments seek to answer the question “does the model know when dynamics along its own generated trajectories are invalid?”.
> Regarding the differences between Fig 26 and 27, you are absolutely correct. We believe this provides novel intuition over how dynamic modelling errors arise during a rollout.
>
> **“Displacement Variable”:**
> Yes, you are correct. This sets the location of the agent in the MuJoCo simulator, but the transition function at all locations is exactly the same, so we leave this value fixed, as all we care about is the transition function $T(s’|s,a)$, and the displacement variable doesn’t form part of the state that is fed into the agent.
>
> **“Evaluation of performance in real environments”:**
> We address this point in our general response. However, we’d like to highlight that new results in our general response now use a single hyperparameter setup, which shows the general applicability of our findings and recommendations with far less online validation required than previous algorithms.
>
> **“Explain ‘probability of improvement’”:**
> The methodology and definition is derived from [1]. RLiable advocates for computing aggregate performance statistics and probability of improvement across many tasks in a benchmark suite. This is important when the number of tasks become large, and also prevents outliers from dominating mean statistics. For normalization, we use the standard D4RL return scaling. In the interests of being self-contained, we have summarized this method in the appendix for the updated draft in the Appendix. Thank you for pointing this out.

---

> > ### Author Response · Authors · 2021-11-18
> > **Response to reviewer PM7Z (Part 2)**
> >
> > **“Log-likelihood v.s. MSE?”:**
> > We reproduce the Transfer experiments from Table 1 in Appendix A.4, except this time measuring correlation to the negative log-likelihood (NLL) of the data under the world model. We observe that overall the correlation is stronger, but this is to be expected as NLL is measured under the aggregation of all elements in the ensemble, thus deriving a measure that is, by construction, more strongly correlated with the overall model uncertainty. This is in contrast with our MSE measure, which compares the MSE under the actual sampling scheme used during actual imaginary rollout (i.e., subsampling an ensemble member, then sampling from its Gaussian posterior), thus it is possible to sample next states that lie near at the long tails of the aggregate (either due to selecting a single ensemble member whose mean lies at these extrema, or just by virtue of randomly sampling such a value). This means it is possible to sample next states that lie far from the model mean whilst that model is confident, thus explaining why MSE shows overall poorer correlation under our Spearman and Pearson statistics. In summary, the MSE correlation can be seen as being more representative of the actual errors encountered during policy training, whereas the NLL correlation can be seen as answering the question of “how well do the penalties capture when the model believes the data is unlikely?”.
> >
> > **“Minor comments”:**
> > Thank you for pointing out these formatting errors, we have addressed them in the updated manuscript. Regarding the use of max aleatoric in MOPO, the authors justify this empirically in their Appendix D, showing that an epistemic uncertainty measure, over their limited tuning, appears to perform worse, and that max aleatoric performs best. We believe our work questions this conclusion, as validated by the strong performance of principled penalties that explicitly incorporate both uncertainties.
> >
> > [1] “Deep reinforcement learning at the edge of the statistical precipice”, Agarwal et al., NeurIPS 2021

---

> > ### Comment · Reviewer_PM7Z · 2021-11-19
> > **Clarification about rollout error calculations and accumulation of errors**
> >
> > First, thank you very much for the detailled response to all my comments.
> >
> > I would just like a clarification about the rollout error calculations and especially its connection with error accumulations. You wrote : "What we validate is whether the predicted next state at each timestep is correct under the simulator (which we do by resetting the state of the simulator to the states predicted under the world model at each time step)". However in the paper it is written that "as highlighted in Fig. 1a, errors do not always accumulate during a single rollout in the model.". If the state of the simulator is reset to the state predicted by the model at each time step I do not think you can say that this is related to error accumulation. To judge about error accumulation you would need to start from the same $s_0$ for both the model and the simulator, the model generating $\hat s_1, ..., \hat s_T$ where for each $t$ we have $\hat s_{t+1} = \hat f(\hat s_t, a_t)$, with $\hat f$ the model, and the simulator (replaying the same actions) generating $s_1, ..., s_T$ using $s_{t+1} = f(s_t, a_t)$. Then you would compute the error between $\hat s_t$ and $s_t$ for each time step. Whereas "resetting the state of the simulator to the states predicted under the world model at each time step" would mean that you use $s_{t+1} = f(\hat s_t, a_t)$ to generate $s_1, ..., s_T$.

---

> > > ### Author Response · Authors · 2021-11-20
> > > **Reply to point on rollout error and accumulation of errors**
> > >
> > > We thank the reviewer for the request for further clarification. We believe these are two separate phenomena, which arise along a model generated trajectory:
> > >
> > > 1. Error in dynamics from modelling the transition function (which we investigate in our paper)
> > > 2. State drift away from the real environment given the same sequence of actions (the setting you describe)
> > >
> > > The latter phenomenon has been investigated in the past [1,2] showing that indeed, errors can accumulate over time resulting in significant state drift. These papers argue that this type of accumulation prevents rolling out the model for long horizons to learn useful policies. Hence, they use off-policy methods with 1-step TD learning and short horizons to ameliorate this.
> > >
> > > However, empirical evidence counters this and shows that it is in fact possible to use long rollouts; hence we design this experiment to explain why. We show that despite potential accumulated state-drift, the actual transition function learned by the model appears to generalize. Concretely, whilst the state-action pair fed to the model may actually lie very far from the offline data, the model is still able to generate faithful transition dynamics as measured under the true environment. Crucial to this is that we accurately estimate dynamics error and down-weight reward accordingly. As we state in the paper, we hypothesize this could partly explain the effectiveness of model-based methods in offline RL, as they can effectively generate transition tuples that may aid with robustness through data augmentation, which has recently shown to help significantly with policy learning [3,4].
> > >
> > > [1] [When to Trust Your Model: Model-Based Policy Optimization](https://arxiv.org/abs/1906.08253), Janner et al., NeurIPS 2019
> > >
> > > [2] [Offline Reinforcement Learning as One Big Sequence Modeling Problem](https://arxiv.org/abs/2106.02039), Janner et al., NeurIPS 2021
> > >
> > > [3] [Reinforcement Learning with Augmented Data](https://arxiv.org/abs/2004.14990), Laskin et al., NeurIPS 2020
> > >
> > > [4] [Augmented World Models Facilitate Zero-Shot Dynamics Generalization From a Single Offline Environment](https://arxiv.org/abs/2104.05632), Ball et al., ICML 2021

---

> > > > ### Comment · Reviewer_PM7Z · 2021-11-22
> > > > **Reply**
> > > >
> > > > Thank your for the clarification.
> > > >
> > > > I think it would be better to make this point clear in the paper. I guess I was confused by the use of the term "error accumulation" in the sentence ""as highlighted in Fig. 1a, errors do not always accumulate during a single rollout in the model". Is it actually correct to talk about error accumulation when showing single step errors by resetting the state of the real system?
> > > >
> > > > You wrote in your reply "despite potential accumulated state-drift" and "whilst the state-action pair fed to the model may actually lie very far from the offline data": how do we know from the experiments that we are actually far from the offline data?
> > > > It is possible that the model "wrongly" predicts a state $s_{t+1}$ that makes the system return to a known region of the states (from which lots of samples were collected in the offline data) and therefore the next single step prediction will be good, however the whole trajectory prediction is bad.
> > > >
> > > > I however agree with the following sentence "Crucial to this is that we accurately estimate dynamics error and down-weight reward accordingly" as the uncertainty estimation looks correlated to the dynamics errors for each single step.

---

> > > > > ### Author Response · Authors · 2021-11-23
> > > > > **Thank you for your reply**
> > > > >
> > > > > Thanks very much for further clarifying and for pointing that the source of confusion. On the two points:
> > > > >
> > > > > **Is it actually correct to talk about error accumulation when showing single step errors by resetting the state of the real system?**
> > > > > We agree that the terminology we use could definitely cause confusion, and have updated the manuscript to make the distinction between state-drift and transition-error clearer. We will change that sentence to reflect that: dynamics errors do not necessarily increase as we consider longer imagined trajectories.
> > > > >
> > > > > **How do we know from the experiments that we are actually far from the offline data?**
> > > > > This is an important question - to determine whether model generated trajectories drift from the offline dataset, we measure the L2 distance between the generated state-action tuples and the closest data point in the offline training set. These results are shown as the red lines in Figures 26 and 27 of the Appendix. We observe that for most environments, immediately after a “dynamics spike”, the data becomes more out-of-distribution as measured by L2 distance. However, whilst dynamics error may recover in the immediate next step, we can enter a region of drift away from the original data; we clarify this point in Figure 28. We are not the only authors to use state-action L2 distance in the proprioceptive setting; in [1] the authors successfully use this distance to facilitate imitation learning.
> > > > >
> > > > > Furthermore, in Appendix D.3, we also generated a t-SNE plot of imagined trajectories vs. a mixed dataset on the Hopper environment. We find that there is some overlap of the imagined trajectories with the D4RL data, but that the imagined trajectories also reside in parts of the manifold where there is little coverage from the D4RL data.
> > > > >
> > > > > We have updated our manuscript to better incorporate the above discussion.
> > > > >
> > > > > ----
> > > > > [1] Primal Wasserstein Imitation Learning, Dadashi et al., ICLR 2021 [OpenReview](https://openreview.net/forum?id=TtYSU29zgR)

---

> > > > > > ### Comment · Reviewer_PM7Z · 2021-11-23
> > > > > > **Thank you**
> > > > > >
> > > > > > Thank you for the clarifications.

---

### Official Review · Reviewer_eM3K · 2021-10-31

**Correctness:** 4
**Technical Novelty And Significance:** 2
**Empirical Novelty And Significance:** 3
**Recommendation:** 6
**Confidence:** 4

**Main Review:**

The main contribution of this paper is to systematically compare several uncertainty estimate methods. Perhaps surprisingly, such comparison has not been conducted before, even in those papers that originally proposes to implement a pessimistic MDP. To this end, this paper fills an important gap in the literature. The paper is also well written. The experiment setups are clearly defined.

However, my major concern is that the technical contribution of this paper is not strong enough. Other than designing several interesting experiments to compare model uncertainty estimation methods, the paper does not contribute enough new insights that can potentially be interesting for a broader research community.

Furthermore, I think a very related paper is missing (Abbas et al., 2020). This paper also compare several uncertainty estimation methods in model based RL, including the ensemble based approaches (Lakshminarayanan et al., 2017). The main conclusion is very similar: a combination of epistemic and aleatoric uncertainty is the best choice.

Abbas, Zaheer, et al. "Selective Dyna-style planning under limited model capacity." International Conference on Machine Learning. PMLR, 2020.



**Summary Of The Paper:**

Model-based offline reinforcement learning algorithms typically involve constructing a pessimistic MDP, which is implemented based on an uncertainty estimation of the learned model. This paper conducts empirical analysis to compare different design choices of the uncertainty estimation in practice. In more details, the authors compare different approaches in terms of the correlation between the estimated uncertainty and ground truth model error. They also use bayesian optimization to search the best hyperparameter configuration that achieves strong empirical performance.

**Summary Of The Review:**

I recommend rejecting this paper because the technical contribution is not strong enough.

---

> ### Author Response · Authors · 2021-11-18
> **Response to reviewer eM3K**
>
> Thank you for your review, we were very encouraged to see you agreed that such work is required currently for offline MBRL, and could see merit in our experimental approach.
> Thank you for your detailed and thoughtful review. We hope this response addresses the concerns and questions raised in your review. Please let us know if any other issues remain.
>
> **“Interest to broader research community”:**
> We have addressed this point in the general response, showing that our insights can be readily applied by offline RL practitioners. The setup displayed follows our general recommendations: using the ensemble standard deviation metric and a longer horizon, in a single hyperparameter setup. Thus, our work represents important and actionable guidance.
>
> **“Related work”:**
> Thank you for highlighting this work, which we were previously unaware of. We agree it is highly related, despite featuring a rather different problem and experimental setting (e.g., online RL using MVE, limited small environments, max horizon of 4). We are pleased to see they tackle a very related question, namely how to incorporate model uncertainty into planning, and draw similar conclusions with one of our key findings. We have added this citation to our paper and to the related work.

---

> > ### Comment · Reviewer_eM3K · 2021-11-24
> > **Response**
> >
> > Thanks for your response. Although the paper is not strong on the technical side, this paper does contribute valuable insights for the offline RL community, especially for practitioners. I have increased my score accordingly.

---

> > > ### Author Response · Authors · 2021-11-25
> > > **Thank you for your response**
> > >
> > > Thank you for your time reviewing our response and for your suggestions which improved our paper.

---

### Official Review · Reviewer_uPzV · 2021-11-02

**Correctness:** 4
**Technical Novelty And Significance:** 2
**Empirical Novelty And Significance:** 3
**Recommendation:** 8
**Confidence:** 3

**Main Review:**

Strong points:


This is a quite thorough evaluation which compares several of the recent popular offline MBRL works, and investigates critical design choices made in each of them. I feel that this is a valuable contribution to research in offline RL, since it’s generally quite difficult to understand where performance improvements between various methods come from due to so many differences in hyperparameters.
The comparison metrics for the uncertainty penalty are quite insightful and reasonable.

Weak points:


The “optimized” version presented in the paper only optimizes and presents statistically significant results compared to MOPO, so it is unclear if similar trends hold for the other methods compared to in the paper and there may not be sufficient evidence to argue that hyperparameter selection affords large improvements for other algorithms.


The “Optimized” numbers are reported by optimizing over each D4RL task. With that, it seems unsurprising that with a single setting and an optimization run over a single seed, the hyperparameters can be tuned better than the original implementations. It seems important to this evaluation to also determine how stable these hyperparameter choices are across random seeds.

Questions/clarifications/nitpicks:


It’s a bit confusing to me to present “optimized” as “our method” in section 6, when the difference (as far as I am aware) is in the hyperparameter selection of MOPO (which I think could be made more clear).

Additionally, in Figure 1, “Optimized” has not yet been introduced, so while in hindsight it is clear that it’s an optimized version of MOPO, it may not be immediately obvious.

Figure 1(b) and the methodology used in this claim is not clearly described in the text


**Summary Of The Paper:**

The paper provides an evaluation of many of the design choices and hyperparameter decisions made in offline model-based reinforcement learning methods which have emerged recently. Particularly, the empirical study looks at uncertainty penalties used in these methods, as well as hyperparameters such as ensemble size, penalty weighting, and rollout horizon. The authors find that offline MBRL methods are quite sensitive to each of these parameters. They compare the “optimized” version of MOPO with hyperparameters tuned using Bayesian optimization, and find that it leads to statistically significant performance improvements over the version of MOPO presented in the original paper.

**Summary Of The Review:**

I recommend accepting this paper. I think that more rigorous evaluations of the design and choices made in RL and particularly offline RL methods are important contributions to help drive future research. I think the analysis of the uncertainty quantification methods is particularly interesting.

---

> ### Author Response · Authors · 2021-11-18
> **Response to reviewer uPzV**
>
> Thank you for your positive feedback, we are very pleased to see that you found our approach thorough, and that you agreed our work represents a valuable step in disentangling algorithmic contributions from hyperparameter choices. We hope this response addresses the concerns and questions raised in your review. Please let us know if any other issues remain.
>
> **“Unclear if this work generalizes to other offline MBRL algorithms”:**
> While we focus our attention on MOPO, we believe that MOPO represents a very general testbed that incorporates design choices common to *many* model-based algorithms, such as model ensembles, Dyna-style planning, incorporation of model uncertainty into this planning, and deep policy-based control (e.g., SAC). At the time of writing this paper, there were not many publically available offline MBRL algorithms (indeed we chose to implement MOPO from scratch in the interests of reproducibility), and those that are available have issues over their [reproducibility](https://github.com/aravindr93/mjrl/issues/42). Therefore, we view the robustness and generality of MOPO as a valid compromise. Furthermore, previous works seeking to understand specific elements of the MBRL framework have also focused on ablations to a single algorithm [1].
>
> **“Found solutions are only over one seed”:**
> We use one seed during the *discovery phase* for optimal hyperparameters under Bayesian optimization, but all results reported are over 4 seeds and averaged over 10 policy training epochs. Thus, all the final reported scores are robust to variation across seeds. We provide more details of the training methodology in our general response.
>
> **“Clarity around a) ‘our method’; b) ‘optimized’; c) methodology in Fig.1b”:**
> On a), we agree this wording is potentially confusing, and will change this to “our approach”. On b), we will clarify what we mean by optimized in the caption. On point c), the methodology is highlighted in the listed paper (i.e., [2]), but we agree that in the interests of being self-contained, we will add details in the appendix about how this was produced.
>
> [1] “On the role of planning in model-based deep reinforcement learning”, Hamrick et al., ICLR 2021
>
> [2] “Deep reinforcement learning at the edge of the statistical precipice”, Agarwal et al., NeurIPS 2021

---

> > ### Comment · Reviewer_uPzV · 2021-11-24
> > **Response to authors**
> >
> > Thank you for the clarifications and justifications provided. The results using the single hyperparameter for all environments provided in the general response particularly help demonstrate the impact of this work for those who use offline MBRL methods in practice. I think this is a valuable contribution, and will raise my score.

---

> > > ### Author Response · Authors · 2021-11-25
> > > **Thank you for your response**
> > >
> > > Thank you for your time reviewing our additional results and for your comments which made our paper better.

---

### Official Review · Reviewer_TWSZ · 2021-11-03

**Correctness:** 2
**Technical Novelty And Significance:** 2
**Empirical Novelty And Significance:** 3
**Recommendation:** 6
**Confidence:** 4

**Main Review:**

Overall, this work is well written and presents a large range of empirical results on the utility of different uncertainty heuristics. The wide scope of these experiments, evaluating the uncertainty penalties in relation to both model prediction error and downstream MBRL performance is a strength of the approach. The approach highlights that the predictive variance or standard deviation of the ensemble predictions can perform well as an uncertainty penalty, even though past work in offline MBRL has not considered this. In addition, it highlights the important role that hyperparameters play in controlling the performance of the MOPO algorithm.

My main concern with this work is that does not make any actionable claims. They claim that they achieve SOTA results by choosing different hyperparameters than previous work, but this hyperparameter selection is chosen via Bayesian optimization that is specific to the same environment. To me, this is the equivalent of "training on the test set," in that it uses interaction with the true system to select hyperparameters for the same system. In that sense, the result can no longer be called offline RL. Of course, hyperparameter selection is an under addressed topic in the community, and there are many papers which manually select hyperparameters that yield good performance. The fact that explicitly optimizing these hyperparameters through interaction on the test environment can yield better performance is not very surprising, but also not very useful for the target applications of offline RL where interaction with the true system is expensive or safety-critical. Indeed, these results do not even paint a consistent picture where one uncertainty penalty is clearly preferred to the others. The work would be significantly improved if the authors used the results of their empirical experimentation to propose a general recommendation for penalty selection and hyperparameter selection.

Other comments:
- Experiments are limited to deterministic environments where the distinction between aleatoric and epistemic uncertainty is unimportant. In stochastic environments, I don't see why aleatoric uncertainty should factor into a reward penalty, so the ensemble variance and standard deviation penalties proposed may not work well in such environments.
- The claim that the ensemble standard deviation or variance is "arguably the most principled" choice is not supported. Why is the variance of the ensemble predictions match the total variation distance term? What makes it "more closely aligned with the theory"?
- The AUC / AP are hard to compare across percentile choices because the relative proportion of positive and negative samples is also changing, so the AUC and AP of a random classifier changes across this task. It would be interesting to report these results on a subsample with even proportions of low-error and high-error samples such that the AUC of a random classifier would be 0.5 in all cases. Furthermore, the precision recall curves in the appendix don't make sense to me, as recall should vary all the way from 0 to 1 as the threshold changes. Perhaps the axes are flipped?
- It is unclear what is meant by "latent dynamics that are KL-regularized to a spherical Gaussian" (last paragraph of section 4.2)

**Summary Of The Paper:**

The authors present an empirical study of several uncertainty quantification heuristics applied to model learning in offline model-based reinforcement learning. Specifically, they consider the basic architecture of MOPO, in which an uncertainty based state-action penalty function is applied on top of the standard reward to construct a pessimistic MDP. Within this set up, the authors perform a empirical study of several different uncertainty penalties, exploring their correlation to the prediction error of the model, as well as in terms of their ability to detect individual transitions with high-percentile prediction errors. Finally, the authors perform Bayesian optimization over the choice of the uncertainty penalty as well as other hyperparameters such as number of ensemble elements, planning horizon, penalty weights, etc. They present these results showing that the optimal choice of penalty and penalty weight can vary significantly, not only between environments, but also within an environment as a function of offline dataset.

**Summary Of The Review:**

Overall, the paper presents a large amount of empirical results, but is lacking a well supported, clear actionable message beyond using longer rollout horizon and ensemble standard deviation as an uncertainty penalty. Even this claim is lacking empirical support, as there is no evaluation of a fixed, guiding principle for hyperparameter selection that is held constant over a wide range of domains. As such, I'm hesitant to recommend acceptance. If the manuscript is updated to have clearer empirical evidence for the main claims in the paper, including a fair evaluation of the recommended strategy across a variety of domains without optimizing hyperparameters separately for each environment, I would increase my score.

---

> ### Author Response · Authors · 2021-11-18
> **Response to reviewer TWSZ**
>
> Thank you for your helpful feedback, we are encouraged to see that you found our approach thorough and revealing in terms of the importance of hyperparameter choices. We hope that our unified experiments using a single hyperparameter setup assuage your concerns, and we hope this response addresses the remaining questions raised in your review. If this is the case, then we humbly ask you to consider raising your score. If not, please let us know if any other issues remain.
>
> **“[Lack of] actionable claims”:**
> We have addressed this point in the general response, showing that we can achieve impressive results by following our general recommendations: using the ensemble standard deviation metric and a longer horizon, in a single hyperparameter setup. Indeed, these are the best results we know of that are evaluated in this way, and go a long way in applicability of offline RL algorithms to settings where interaction with the true system is expensive or safety-critical. We thank you for raising this issue.
>
> **“What about stochastic environments?”:**
> To our knowledge, none of the methods in the offline MBRL literature have considered stochastic dynamics (indeed none of the environments in D4RL are stochastic). We believe that this is out of the scope of this paper because the impact of risk-sensitivity/uncertainty penalization (especially for rollouts performed inside the model) for even standard deterministic environments (predominant in continuous control) has been poorly understood up to now. In short, we believe it is important to understand the intricacies and pathologies of existing design choices on common environments before delving more deeply into less common benchmarks. We expand further in the general response on the distinction between using “aleatoric” and “epistemic” uncertainty measures, and thank you for raising this point.
>
> **“Is ensemble std/var more aligned with theory?”:**
> The ensemble standard deviation or variance corresponds to a formal metric of uncertainty associated with the ensemble - namely the actual underlying standard deviation or variance of the Gaussian mixture, and is thus most commonly used for uncertainty quantification in Bayesian literature. Empirically, we show that this principled measure in fact also correlates best with the dynamics MSE. We note that MSE is motivated by theory, whereby assuming a Wasserstein form of the IPM (instead of the total variation distance) in Eq 1, we retrieve a $|G|$ that represents the dynamics MSE. Through this lens, the MSE can be considered the ‘oracle’ which may be used to provide an upper bound over the expected return.
>
> **“Axis flipped?”:**
> We greatly thank the reviewer for correctly pointing out the flipped axes in our Precision-Recall curves, and have replaced all plots in the paper with corrected versions.
>
> **“What is meant by KL-regularized latent dynamics?”:** This specifically refers to the modelling choices in LOMPO [1], which considers offline MBRL from pixels. In this work they use the Dreamer RSSM model [2], which optimizes a variational lower bound on a latent recurrent model, where the latent variable aims to model the underlying state (i.e., a latent dynamics model). The resulting loss function includes a term that can be described as a regularization term, enforcing the dynamics posterior distribution to be close to a spherical Gaussian prior.
>
> [1] Offline Reinforcement Learning from Images with Latent Space Models, Rafailov et al., 2020, arXiv:2012.11547
>
> [2] Dream to Control: Learning Behaviors by Latent Imagination, Hafner et al., ICLR 2020, arXiv:1912.01603

---

> > ### Comment · Reviewer_TWSZ · 2021-11-22
> > **Thanks for the reply and inclusion of single hyperparameter selection results**
> >
> > Thanks for the thorough response.
> >
> > The results using a single hyperparameter setting across all domains, in combination with the automatic penalty weighting are impressive and greatly improve the significance of this work by giving concrete evidence to support the main takeaways of this paper. I acknowledge that this hyperparameter tuning is done per environment in prior work, but in spirit this procedure goes against the core idea of offline RL. Indeed, understanding how to tune hyperparamters for offline-RL in a principled manner with zero or limited online interaction represents an important and open question in community. Within this context, showing how a good set of hyperparameter choices can generalize across tasks and improve baseline performance is a good contribution. I will accordingly raise my score.
> >
> > Regarding the other points:
> >
> > *Stochastic Environments*: Thanks for clarifying and updating the manuscript to include references to relevant related work. Given that the notation and many of the penalties are designed to support stochastic dynamics, I would recommend clearly stating (perhaps in the conclusion) that this analysis focuses on deterministic environments, and that the optimal choice of uncertainty measure may be different if applying these results to stochastic environments.
> >
> > *Ensemble std/var theory*: I agree that the ensemble std uncertainty penalty is the standard deviation of the Gaussian mixture model constructed by combining the Gaussian prediction made by each ensemble member, and that this mixture model is an approximation of a Bayesian posterior predictive distribution. However, in order to say that the this penalty is "more principled" in the context of offline RL, you need to make a case for why the standard deviation of this posterior predictive distribution should be more closely related to model mismatch than any other uncertainty metric. Indeed, this is demonstrated through the thorough experiments, but I don't believe this experimental evidence is enough to say the ensemble std/var approach is "more principled" than the other approaches, and would recommend changing the wording.
> >
> > *KL-regularized latent dynamics*: Thanks for providing this clarification. It makes sense that distributional mismatch is less likely to happen in a learned latent space, so using a likelihood based technique may be better suited to learned latent space dynamics. I still find the language in the manuscript a bit confusing here, perhaps you could update it to make it more clear that this penalty was designed for latent space dynamics which can be learned to have well-behaved dynamics.

---

> > > ### Author Response · Authors · 2021-11-23
> > > **Thank you for your response**
> > >
> > > Thank you for your response, and we are happy that you believe the single hyperparameter experiments showed our broad recommendations are generally applicable to offline RL practitioners. Indeed, fully offline policy evaluation is an open question in this field, and is crucial to “truly offline” RL. We hope that our experiment, nonetheless, showed a more realistic application of offline RL. Regarding the other points:
> > >
> > > **Stochastic Environments:**
> > > We agree this is an important point to clarify - we have now made it clear in our conclusion that we focused on deterministic environments. Optimal choice of uncertainty penalty in stochastic environments is an underexplored area in offline RL, and indeed many of the penalties we investigate (particularly the ensemble penalties that we settle on) support stochastic dynamics. We believe the novel methods of analysis in our work will be just as relevant in this setting.
> > >
> > > **Ensemble std/var theory:**
> > > We agree that our argument was heavily supported by strong empirical evidence that the ensemble std correlates far better with the MSE dynamics error, which provides a theoretical lower bound on the reward. We have updated the draft by referring to the ensemble penalties as more ‘canonical’ choices, and have justified the use of this description in the manuscript. Thank you for pointing this out.
> > >
> > > **KL-regularized latent dynamics:**
> > > We thank you for the suggestion - we have updated our manuscript to make it clear that the penalty was designed for latent space dynamics constrained to be well-behaved:
> > > > ...it was designed for use with a KL-regularized latent state space model which has well-behaved dynamics.
> > >
> > > We thank you again for your suggestions which have improved our paper.

---

### Official Review · Reviewer_DXYY · 2021-11-03

**Correctness:** 3
**Technical Novelty And Significance:** 3
**Empirical Novelty And Significance:** 3
**Recommendation:** 6
**Confidence:** 4

**Details Of Ethics Concerns:**

Not applicable.

**Main Review:**

I think the research is well-motivated and interesting, while I have some concerns about the experiments. I give the comments in the following.

Pros:

1. The research is well-motivated.  The gap between the true model and the learned model is estimated by uncertainty measurements. Nevertheless, in the MOPO and Morel papers, different uncertainties are not rigorously studied. I think this problem is important and should be studied in-depth.

2. The use of Spearman rank and Pearson bivariate to identify the correlation between uncertainty and true MSE is novel. The experiments verify such measurements are closely related to the performance.

Cons:

1. The experiments in section 4.2 are a bit confusing. Why should we use exploitative policies that are generated by non-penalty methods to generate trajectories? The non-penalty methods cause overestimation and poor performance, and the resulting policy may be similar to a random policy. The trajectories generated by such policies may lie in a small range of state-action space. I suggest using the trajectories from the medium-replay dataset to evaluate since it covers a large state-action space. You can point it out if I don't understand correctly.

2. The aim of section 5.3 is to study the effect of rollout horizon. It is unclear how the results presented in table 2 relate back to the rollout horizon, since I do not find the rollout horizon is used as a variable in Table 2. The dynamics discrepancy should be related to the horizon length. Can you explain more?

3. In Table 3, the results mix the D4RL v0 and v2 datasets. It is not valid since these two datasets are quite different. The v2 dataset is much larger than the v0 dataset. I suggest unifying all results in a single dataset. Maybe you can refer to other offline RL papers released recently to get the scores of MOPO and Morel in v2 dataset.

4. Considering different tasks have different optimal hyper-parameters, I suggest adding additional analysis about the sensitiveness of N, lambda, and h in Table 3. E.g. Should lambda be larger in single-policy datasets and be smaller in mixed datasets?  Does a larger N perform better? Does N affect the performance significantly if N>10? Maybe h is sensitive to different tasks?

5. The author should explain why “Ensemble Standard Deviation” measures both the epistemic uncertainty and aleatory uncertainty. The paper should be self-contained. What about other methods considered in section 4 (in measuring the epistemic uncertainty and aleatory uncertainty)? Can we draw the conclusion that both the aleatoric and epistemic are important to offline RL?

Minor:

1. The method to obtain true dynamics error should be given in discussing Figure.1.  Although it has been discussed later in the paper.

2. At the end of section 5.1, the paper writes “This again suggests that the number of models not only affects the quality of the estimation, but also its distributional shape. ” Can you explain more?

**Summary Of The Paper:**

The paper provides detailed analysis of different uncertainty quantifications in Model-based offline RL, from both statistical and empirical perspectives. Further, the paper performs Bayesian optimization to find hyper-parameters and the optimized hyper-parameters perform better empirically.


**Summary Of The Review:**

The paper is well motivated while it needs to be improved in clarity. (1) The dataset used in evaluation should be unified. (2) The experimental setup in Table 1 and Table 2 should be explained more clearly. (3) Additional ablation or explanation of Table 3 is suggested to be added. (4) Some discussion about the epistemic and aleatoric uncertainty.

---

> ### Author Response · Authors · 2021-11-18
> **Response to reviewer DXYY (Part 1)**
>
> Thank you for your detailed and thoughtful review. We were pleased to see that you believe our work is both well motivated and novel. We hope this response addresses the concerns and questions raised in your review. Please let us know if any other issues remain.
>
> **“Why use...exploitative policies?”:**
> One of the primary failure modes of offline MBRL and MBRL in general is model-exploitation due to model misestimation. Thus, we train policies that are designed to exploit the model (over-estimate expected return compared to the real world) specifically to investigate this phenomenon. This allows us to better understand how well existing penalties can detect when this occurs. This problem is particularly important in offline MBRL as we are unable to sample new real trajectories that can correct for any model misestimation. We will update our discussion to reflect this.
>
> Regarding diversity of the data, we thank you for raising this very interesting question. One important thing to note is we gather exploitative trajectories from policies deployed *inside the world model*, not in the actual environment, as we are seeking to measure the calibration of our penalties under the covariate shift experienced by planning inside the model with these experiments. However, we fully accept that the issue of diversity still applies to policies deployed inside the world model, so we have performed additional analysis in Appendix D.3, using the suggestion of comparing to the medium-replay data. In short, we find that the induced trajectories appear to be suitably diverse under our training methodology, and also show coverage of exploitative regions outside the training data. In addition to this, we provide the individual medium-replay results in the following table as per your request, showing the average performance over all offline datasets transferring to the Medium-Replay data, with +/- 1 standard deviation over 3 model seeds:
>
> |                    |     Hopper    |     Hopper    |  HalfCheetah  |  HalfCheetah  |
> |--------------------|:-------------:|:-------------:|:-------------:|:-------------:|
> | Penalty            |     $\rho$    |      $r$      |     $\rho$    |      $r$      |
> | Max Aleatoric      | 0.74 +/- 0.00 | 0.54 +/- 0.01 | 0.75 +/- 0.01 | 0.45 +/- 0.01 |
> | Max Pairwise Diff. | 0.76 +/- 0.00 | 0.61 +/- 0.01 | 0.78 +/- 0.00 | 0.61 +/- 0.02 |
> | Ensemble Std.      | 0.79 +/- 0.00 | 0.63 +/- 0.01 | 0.80 +/- 0.00 | 0.61 +/- 0.01 |
> | Ensemble Var.      | 0.79 +/- 0.00 | 0.66 +/- 0.01 | 0.80 +/- 0.00 | 0.64 +/- 0.01 |
> | LL Var.            | 0.05 +/- 0.04 | 0.08 +/- 0.02 | 0.29 +/- 0.03 | 0.13 +/- 0.02 |
> | LOO KL             | 0.01 +/- 0.04 | 0.02 +/- 0.03 | 0.05 +/- 0.03 | 0.08 +/- 0.01 |
>
> As we see, the trends here are virtually identical to those over the entire D4RL dataset (i.e.,Table 1). We can provide individual transfer statistics too if that would be of interest (e.g., Random -> Medium-Replay).
>
> **“Table 2 relation to horizon unclear”:**
> The analysis in the table is to precisely identify the effectiveness of our penalties at capturing the observed “dynamic error spikes” that occur under longer horizons. A key reason why this hyperparameter is often left very small (in MOPO the horizon does not exceed 5) is precisely to prevent the manifestation of these spikes. However, our table shows that certain penalties capture these spikes very well, giving us confidence that we can extend horizons to significantly larger values, which goes against conventional wisdom. We validate this finding empirically, and subsequently show that we almost always improve performance by increasing horizon due to the increase in valuable on-policy information to train the policy.

---

> > ### Author Response · Authors · 2021-11-18
> > **Response to reviewer DXYY (Part 2)**
> >
> > **“Dataset comparison is not valid”:**
> > We believe our comparison is fair, as *our own algorithms are only ever trained on the harder v0 dataset* (see the bottom of Page 8 [here](https://arxiv.org/pdf/2110.06169.pdf) and this [GitHub Issue](https://github.com/rail-berkeley/d4rl/issues/86) for more discussion about v0 v.s. v2). The COMBO algorithm also uses the original v0 dataset but unfortunately their code has not been made publicly available to unify the results. We believe **the anomaly lies with the MOReL results which they present using the v2 datasets**; we also have emphasized this in the table caption. In the MOReL paper, they in fact compare against other v0 results, despite training on the easier v2.
> >
> > Thus, our competitive results further validate our analysis, given that we are likely to be penalizing our own approach as compared to MOReL. We also restate that the aim of our paper is to provide deeper understanding and analysis of key design choices in offline MBRL, not to show that we obtain SoTA results; the latter is merely a nice by-product that validates our prior analysis, and proves that these design choices are vital for strong performance in offline MBRL. Note that all other hyperparameters (e.g., learning rate, weight decay, network architecture) are kept constant.
> >
> > **“Analyze sensitivity of hyperparameters”:**
> > Our experiments in the general response allow us to further answer this question. They show that when we fix a specific penalty type (e.g. ensemble standard deviation), the Walker2d and HalfCheetah environments prefer a relatively small value for lambda (around 2-5) whereas the Hopper environments prefer higher (around 30-50). Our existing results in Table 3 show that in general, Hopper and Walker2d prefers higher rollout horizons than HalfCheetah.
> >
> > We may see in Figure 3, the correlation to the true dynamics error does not change much with changing N. Together with the stability of the ensemble penalties to changing values of N, we may expect this hyperparameter to be less sensitive than the previous two, Nevertheless, the shape of the distribution does change as shown in Figure 2 and Appendix C.2,  so this hyperparameter still warrants attention. We will include this and further analysis in the updated manuscript.
> >
> > **“Explain how we capture both epistemic and aleatoric uncertainty”:**
> > We follow [1] when calculating our variance and standard deviation terms, observing that these form the aggregate over a uniform mixture of Gaussians (where each component corresponds to an ensemble member). The two components of the variance may be separately interpreted: the portion computing the difference away from the mean is commonly considered the epistemic uncertainty, whereas the average over the variance heads accounts for the overall aleatoric uncertainty.
> >
> > **“Explain more about distributional shape point in 5.1”:**
> > This observation concerns the apparent paradox of improving Spearman rank correlation with increasing model count whilst simultaneously decreasing Pearson bivariate correlation. The way to resolve this is to note that fewer models give a penalty that is more *linearly correlated* with respect to the ground truth MSE, but less rank correlated. This implies the shape statistics (e.g., skew/kurtosis) must be affected by model number, as increasing model number induces a non-linear relationship.
> >
> > [1] Simple and Scalable Predictive Uncertainty Estimation using Deep Ensembles, Lakshminarayanan et al., NIPS 2017

---

> > > ### Comment · Reviewer_DXYY · 2021-11-23
> > > **Thanks for the reply**
> > >
> > > Thank you for your response. I am happy to see more experiments in more diverse datasets. The other clarifications are reasonable. I have increased my score accordingly.

---

### Author Response · Authors · 2021-11-18
**General Response (Part 1)**

We’d like to thank all the reviewers for their time and detailed feedback. We were encouraged that reviewers believed our work was well-motivated and fills an important gap in the literature. Reviewers further agreed that our work presented useful and novel insights into offline model-based reinforcement learning.

We identified a few common themes amongst reviewers, and will address them first in this reply, before moving onto individual responses. We are also very happy to present and discuss a competitive set of results using *a single hyperparameter setup for all environments* which demonstrate the broader applicability of the key takeaways presented in the paper. We also wish to draw attention to the appendix, to which we have added more experimental details, results, and illustrative experiments.

We greatly appreciated the reviewers’ questions and comments, which will greatly improve the updated paper. We have made an effort to provide all the requested analysis, including the results below; we therefore humbly request that the reviewers consider raising their scores.

**Actionability of the insights presented in the paper (Reviewers: TWSZ, eM3K):**
A common theme amongst reviewers was how readily one may apply the conclusions drawn in the paper due to the diversity of the solutions we found and our method of evaluation using BO. We strongly believe that there are numerous interesting and actionable insights to be gained from our work, and this sentiment is reflected by several reviewers. We will first discuss the insights already presented in the work, and then discuss our new results in the next point:

* To our knowledge, we are the first authors to compare a number of published uncertainty penalties on the same offline testbed. We found that despite the apparent utility of various heuristics, standard uncertainty measures ultimately perform best under both (MBRL-focused) calibration and empirical performance. Indeed, researchers have already begun applying the penalties that we analyze onto novel algorithms, despite us showing their choices are likely suboptimal compared to alternatives [1]. *Going forward, our work should encourage practitioners to adopt more accurate reward penalties when choosing how to approximate model uncertainty, a key design choice in many model-based applications.*
* For the first time, we show that *long-horizon model rollouts are effective for offline planning*, which goes against conventional wisdom [2,3]. We design a novel experiment to explain why this is the case, and then empirically validate this analysis by showing statistically significant improvements over shorter horizon approaches. Thus, practitioners should feel confident considering similarly long horizons for offline rollouts.
* Finally, we believe that we are the first to highlight the importance of penalty shape, and show that despite certain penalties showing unfavorable calibration/theoretical underpinnings (e.g., max aleatoric), they can still be selected by our BO algorithm as they display extreme kurtosis and skew statistics. This should pave the way towards research which considers meta-learning parameters to control these higher order statistics, as well as seek to understand why shape matters from a reward-shaping perspective.

**Results obtained using a single hyperparameter setup (Reviewers: TWSZ, uPzV, eM3K):**
We further demonstrate the actionability of our insights by presenting a new experiment where we implement our recommended strategy across the entire MuJoCo Offline benchmark using *a single hyperparameter setup* for all environments.

|                                            Algorithm                                            | Average D4RL MuJoCo Score | Probability of Improvement over MOPO |
|:-----------------------------------------------------------------------------------------------:|:-------------------------:|:------------------------------------:|
| MOPO (default hyperparameters)                                                                  |            34.2           |                   -                  |
| Single hyperparameter setup:  (horizon = 20, constraint = 1)                                    |        49.0 (+43%)        |                73.96%                |
| Choice of two setups:  argmax{(horizon = 10, constraint = 0.5), (horizon = 20, constraint = 1)} |        57.8 (+69%)        |                80.20%                |
| Optimized MOPO (ours, Table 3)                                                                  |        65.2 (+91%)        |                89.06%                |

---

> ### Author Response · Authors · 2021-11-18
> **General Response (Part 2)**
>
> We only use the ensemble standard deviation penalty (a principled yet simple pre-existing measure), 10 ensemble members, and a rollout horizon of 20 (much larger than previous) and automatically tune the penalty weight on-the-fly to a constraint value of 1 (analogous to the SAC entropy weight). In contrast to having to set a penalty weight per environment, *this only involves setting a single hyperparameter across all environments*. With this single set of hyperparameters, we get a 43% boost in performance over the configuration used in MOPO, which was tuned per environment. This clearly shows the insights from our analysis can provide large performance gains more generally.
>
> We’d like to re-emphasize the significance of using only a single hyperparameter setup for obtaining these results. For example, across the 12 D4RL MuJoCo environments, the MOPO algorithm uses 5 different hyperparameter configurations (for penalty weight and horizon). MOReL tunes the following differently for each environment: the negative reward, horizon, number of policy updates, number of trajectories used for gradient calculation, conjugate gradient steps, and network weight initialization. They also tune world model hyperparameters (that we left fixed across all environments), such as when to early stop world model training, as well as its architecture, giving a total of 8 hyperparameters. Finally, COMBO tunes the following hyperparameters per environment: the rollout length, Q-function and policy learning rates, conservative coefficient, choice of sampling distributions.
>
> This restriction shows the strength of our insights and displays the most realistic possible application of offline RL to representative problems. If we were to allow ourselves to take the maximum over just 2 hyperparameter setups still using the ensemble standard deviation (the second setup being horizon=10, uncertainty constraint=0.5), *we are able to obtain an average reward of 57.8, an increase of 69% over MOPO*. This remains below half the number of different configurations used by MOPO, MOReL and COMBO. Finally, as we allow ourselves to tune our hyperparameters to the number of different configurations used by MOPO, MOReL and COMBO, we recover the full performance given in Table 3. We also note that MOReL and COMBO tune over more hyperparameters than we do even in our full Optimized setting. We have included these results in our updated paper and provide a detailed analysis.

---

> > ### Author Response · Authors · 2021-11-18
> > **General Response (Part 3)**
> >
> > **Our results come from training on the test set (Reviewers: uPzV, TWSZ, PM7Z):**
> > We restate that the primary aim of this work is to provide insights into key design choices for offline MBRL, not to obtain state-of-the-art results. Therefore, running BO on the online test environment was a key step in achieving this, as it provides a ‘golden source’ of validation that our lower level analysis in the first part of the paper (concerning: a) penalty calibration; b) horizon length; c) model count) aligns with empirical performance, which we ultimately care about. In other words, we use this approach to confirm our insights over these hyperparameters, and do not claim to introduce a new methodology intended for realistic offline RL.
> >
> > As we note above, most other offline RL will tune hyperparameters per test environment using grid search. This problem is slightly ameliorated by our evaluation protocol, as we optimized specifically for reliability in our BO search. This means we do not require access to the true test environment to select the “best policy” after training fully inside the model. Concretely, and in the interests of transparency, we always report the average performance over the final 10 epochs of policy training (i.e., epochs 990-1,000), which consists of 10,000 SGD steps. To expand on this point, it is well observed that policies can wildly oscillate between optimal and suboptimal across offline training epochs and seeds [such as here](https://github.com/tianheyu927/mopo/issues/5#issuecomment-740388361), so it is impossible to reliably select which policy to deploy without first testing online in these cases; our discovered solutions significantly mitigate this issue.
> >
> > **Why do we care about both epistemic and aleatoric uncertainty? (Reviewers: TWSZ, PM7Z):**
> > We appreciate that the reviewers made the distinction between modelling uncertainty in the transition function for deterministic and stochastic environments. Intuitively, one would believe that in deterministic environments we should only care about modelling epistemic uncertainty, and that aleatoric uncertainty should only be relevant for stochastic environments. However, paradoxically, the original MOPO paper use a penalty solely based on the ‘aleatoric’ portion of the ensemble uncertainty when the D4RL environments are all deterministic. When they performed an ablation based on epistemic uncertainty, this performed worse empirically. This was also observed in work pointed out by another reviewer [4], where they find in practice combining aleatoric and epistemic uncertainties improves performance, despite operating in a deterministic domain.
> >
> > We may explain this by noting: whilst the uncertainty outputs of a deep ensemble may be split into an ‘epistemic’ and an ‘aleatoric’ portion, this is often subject to quality of model fit and the quantity of training data given.  For example, viewing the regression results in [4], ensembling of the aleatoric uncertainty heads appears to have the effect of giving regions with little data support an especially large uncertainty (than compared with only epistemic), providing a stronger signal to the policy to avoid those regions. We know that disentangling epistemic and aleatoric uncertainty is challenging, but believe fully understanding these phenomena would represent very interesting future work.
> >
> > [1] Reset-Free Lifelong Learning with Skill-Space Planning, Lu et al., ICLR 2021, https://arxiv.org/abs/2012.03548
> >
> > [2] When To Trust Your Model, Janner et al., NeurIPS 2019
> >
> > [3] MOPO: Model-based Offline Policy Optimization, Yu et al., NeurIPS 2020
> >
> > [4] Selective Dyna-Style Planning Under Limited Model Capacity, Abbas et al., ICML 2020

---

### Decision · Program_Chairs · 2022-01-20

**Decision:**

Accept (Spotlight)

**Comment:**

This paper empirically studies various design choices in offline model-based RL algorithms, with a focus on MOPO (Model-based Offline Policy Optimization). Among the key design choices is the uncertainty measure used in MOPO that provides an (approximate) lower bound on the performance, the horizon rollout length, and the number of model used in ensemble.

The reviewers are positive about the paper, found the experiments thorough, and the results filling a gap in the current literature. They have raised several issues in their reviews, many of which are addressed in the rebuttal and the revised paper. I would like to recommend acceptance of the paper. Also since the results of this work might be of interest to many researchers working on model-based RL, I also recommend a spotlight presentation for this work.

I have some additional comments:

(1) The paper studies the correlation of uncertainty measures with the next-state MSE, with the aim of showing which one has a higher correlation. The underlying assumption is that the next-state MSE is the gold standard that we should aim for.

If we go back to the MOPO paper, we see that to define an uncertainty-penalized reward, we need an upper bound on the absolute value of G(s, a), which is the difference between the expected value of the value function at the next-state according to the true model and the estimated model.

If we assume that the value function belongs to the Lipschitz function class w.r.t. a metric d, this upper bound is proportional to the 1-Wasserstein distance between the true next-state distributions and the model's distribution.
If the dynamics is deterministic, 1-Wasserstein distance becomes the $d( T(s, a), \hat{T}(s,a) )$. If the distance d is the Euclidean distance, this becomes the squared error.

Therefore, the squared error makes sense for deterministic dynamics, and it only provides an upper bound of $|G(s, a)|$.
If the environment is not deterministic, the squared error may not be a reasonable gold standard anymore to compare the correlation of various uncertainty measures with.

The paper introduces a generic MDP framework, but does not mention anything about its focus on MBRL for deterministic environments until the last sentence of its conclusion. Please clarify this in your camera ready paper.

(2) The experiments are conducted using 3 or 4 seeds. Although this is the common practice in the deep RL community, it is too small. Standard deviations in Tables 1, 2, ... are computed with 3 seeds, which would be cringeworthy to statisticians and empirical scientists. I encourage the authors to increase the number of independent random experiments to make their results more powerful.